# Impact of contaminant size and density on their incorporation into sea ice

Alice Pradel [1] ✉, Rudolf Hufenus [2], Martin Schneebeli [3] &
Denise M. Mitrano [1] ✉

Sea ice accumulates contaminants and redistributes them laterally as the ice drifts and vertically as it melts. Contaminant incorporation into sea ice must be better understood to resolve contaminant cycling and exposure to polar organisms. Here we develop an experimental method that mimics the formation of young sea ice and enables the quantification of model contaminants separately in the ice matrix and brine. Several limitations inherent in field studies are overcome using this approach. Results show that dissolved contaminants (<1 nm) and dispersed colloidal contaminants (1 nm–1 μm) follow the same behavior as sea salts. When colloids aggregate they follow a similar transport pathway to high-density particulate contaminants (>1 μm). While high-density particles are depleted in sea ice and low-density particles are enriched relative to their initial concentration in seawater, both are engulfed and can travel in wide brine channels. These results can also help to predict the incorporation of natural species in sea ice.

There is substantial anthropogenic contamination even in some of the most remote areas on Earth, such as the Arctic Ocean[1–3]. Important fluxes of contaminants are brought to the Arctic Ocean where they can interact with and be incorporated into sea ice, due to the combination of high anthropogenic activity in the Northern hemisphere, patterns of oceanic circulation[4], atmospheric circulation[2,5], and riverine inputs to the Arctic Ocean[6–8]. Once in polar seawater, some contaminants are enriched in sea ice overall[9,10], or are locally enriched within the sea ice porosity[3,11]. For example, microplastic (MPs, 1 μm–5 mm) concentrations were found to be orders of magnitude higher in sea ice compared to the underlying seawater[9,10,12]. MPs concentrations in Arctic sea ice[9,12–14] are in the same order of magnitude as in Arctic sediments[10,15–17], demonstrating that sea ice acts as a temporary sink of MPs. Therefore, the role of sea ice in global contaminant cycling must be assessed, since contaminants in sea ice can be redistributed laterally as sea ice drifts and vertically as it melts[18,19].

Contaminant incorporation into sea ice and their specific location within sea ice must be understood to predict the risks they pose to polar organisms[20,21]. In particular, understanding the magnitude of incorporation from seawater to sea ice is necessary since i) substantial

volumes of seawater freeze annually due climate change (i.e., the replacement of multi-year ice (MYI) by first-year ice (FYI)) and ii) since the seawater/sea ice interface is an important habitat for ice-associated organisms[22,23]. Indeed, microalgae which are the base of the polar food web, thrive in the bottom layer of sea ice due to the unique combination of porosity generated by brine pockets and channels, sunlight and nutrients with seawater[24,25]. This primary productivity could be further enhanced due to climate change, since thin and porous FYI allows for more sunlight and nutrient fluxes[22,25].

The incorporation of contaminants into sea ice during freezing will depend on the freezing process and the contaminants' physico-chemical properties[11,26]. At the onset of freezing, frazil crystals form in the water column and rise to the surface. During this upward movement, they can scavenge particles such as suspended particulate matter[27] and microalgae[28] from the water column and enrich them in bulk sea ice. In contrast, most of the volume of Arctic sea ice is formed by bottom freezing, that is, the vertical advancement of the freezing front in calm conditions, which forms columnar ice. During bottom freezing, dissolved molecules such as sea salts are depleted from the bulk sea ice and accumulated in brine pockets and channels, a process

[1]Institute of Biogeochemistry and Pollutant Dynamics (IBP), Department of Environmental Systems Science, ETH Zürich, Zürich, Switzerland. [2]Empa, Swiss Federal Laboratories for Materials Science and Technology, Laboratory of Advanced Fibers, St. Gallen, Switzerland. [3]WSL Institute for Snow and Avalanche Research SLF, Davos, Switzerland. ✉e-mail: pradel@cerege.fr; denise.mitrano@usys.ethz.ch

known as salt segregation or partitioning[29–31]. This can be followed by gravity drainage, or flushing of the brine which leads to overall desalination of the bulk sea ice[31]. Likewise, dissolved (<1 nm) contaminants such as mercury as well as chlorinated and fluorinated molecules have also been shown to undergo the same transport pathway[3,32,33]. However, the effect of bottom freezing on the behavior of colloidal (1 nm to 1 μm) and particulate (>1 μm) contaminants is less understood[34]. In fact, the accumulation of MPs and nanoplastics (NPs, 1 nm to 1 μm) in bulk sea ice is often attributed to frazil ice without direct evidence[35,36]. Collectively, this highlights our lack of understanding of contaminant incorporation into sea ice.

Field studies have revealed the burden of contaminants in sea ice[1,11]. However, understanding the incorporation process of contaminants during freezing is limited by unknown initial concentrations of contaminants in seawater and the unknown conditions in which sea ice has formed. Furthermore, quantifying some contaminants, such as small MPs (<10 μm) and NPs, in environmental matrices is challenging due to the lack of standardized methods. Some persistent challenges, such as differentiating MPs and NPs from natural organic matter (NOM) and avoiding contamination during sampling and analysis require time-intensive methods. Overall, this limits the amount of samples which can be measured[20,37,38]. For these reasons, we assessed the incorporation of contaminants experimentally under laboratory conditions by generating artificial sea ice in a controlled environment and quantified the incorporation of model contaminants with an array of properties. We hypothesized that the incorporation of contaminants into sea ice would be dictated by their size and, in the case of particulate contaminants, also by their density. On one hand, colloidal species are expected to follow a similar incorporation pathway as dissolved species due to their Brownian motion, which allows them to diffuse away from the freezing front. On the other hand, particulate species with low densities are expected to be enriched in the sea ice relative to the underlying water, due to their positive buoyancy which moves them towards the advancing ice front. Conversely, high-density particles are expected to be depleted by sinking below the advancing ice front. Contrary to dissolved and colloidal species, particles are not expected to be preferentially enriched in the brine but instead may be engulfed by the ice due to the absence of Brownian motion.

To understand the mechanisms of contaminant incorporation into sea ice, we developed an experimental set-up which mimics the growth of columnar sea ice and quantified the incorporation of contaminants into this ice during freezing (Fig. 1). The experimental set-up i) generated environmentally representative artificial sea ice by keeping the salinity of the underlying water constant, and ii) allowed us to quantify a suite of model contaminants within the sea ice. To study the role of contaminant size and density, we used model species with a range of physicochemical properties including a dissolved molecule, two colloidal species (NPs and nanosoot), and four different MPs (two densities and two sizes; Table S1. and Fig. S1). The artificial sea ice formed was harvested to characterize its structure and to quantify species in the underlying liquid, the bulk ice, as well as its two fractions: the solid ice matrix (hereafter called ice) and the brine. This systematic approach allowed us to elucidate how contaminants become incorporated in sea ice. Our results improve our ability to predict how contaminants may be redistributed during sea ice drift and melt and which contaminants organisms living in sea ice will be exposed to.

## Results and Discussion
### Experimental conditions resulted in environmentally realistic artificial sea ice

To assess contaminant incorporation into sea ice experimentally, we were careful to generate environmentally realistic sea ice. This was done by adding artificial seawater (ASW) containing a mixture of humic acid salt and sodium alginate to a column that underwent a vertical temperature gradient of −5.1 ± 0.1 °C to 1.0 ± 0.6 °C, for 19 h (Fig. 2a).

The bottom of the sea ice column was fitted with a flexible tube which extended above the level of the column. This allowed the liquid to move up the tube as the ASW froze and expanded, thereby avoiding pressure increases in the system (Fig. 2, panels a and b). For each experimental condition, three sea ice columns were aligned in series (Fig. 2, panel b). At the end of the experiment, the ice had an mean depth of 3.32 ± 0.61 cm (Fig. 2, panel c). The entire ice cores were lifted out of the column with a solid net attached to two strings and were centrifuged to drain the brine connected to the bottom of the ice core. All species were quantified in the liquid, ice and brine and the structure of the sea ice was assessed by micro-X-ray-tomography (μ-CT). By considering the dimensions of the artificial sea ice columns, the pressure exerted by ice growth and temperature gradients, the

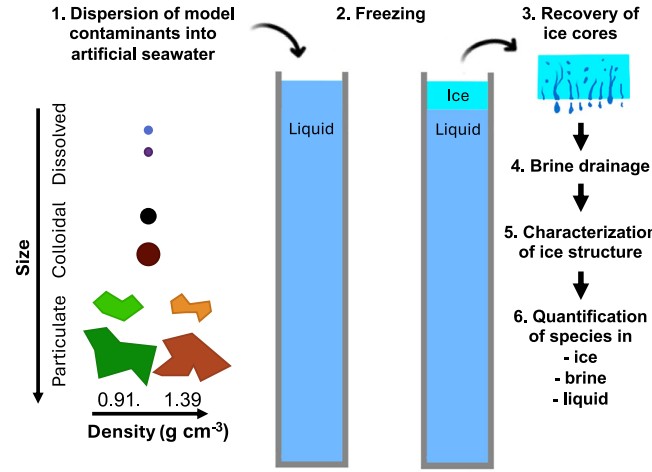

**Fig. 1 | Schematic of the model materials, and workflow of the laboratory grown sea ice analysis.** The experimental design consisted of six steps: 1) For each experiment one type of model contaminant was mixed with artificial seawater and added to columns specifically designed to generate artificial sea ice. 2) Columns underwent controlled freezing conditions. 3) Ice cores were recovered and 4) the brine was drained out of the ice core. 5) The structure of the ice was characterized by micro-X-ray-tomography (μ-CT) and finally 6) salts and contaminants were quantified in the ice, brine and liquid fraction.

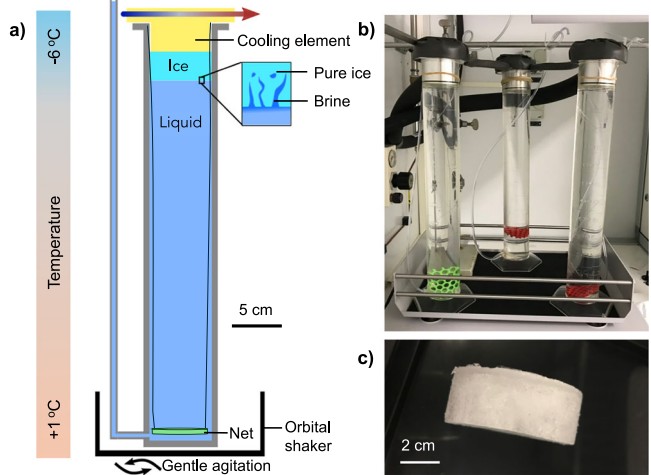

**Fig. 2 | Description of the experimental set-up used to produce artificial sea ice. a** Scheme of experimental set-up used to produce artificial columnar sea ice by bottom freezing. The column dimensions are to scale. **b** Photo of three columns on the orbital shaker before ice production. **c** Photo of an artificial sea ice core after centrifugation.

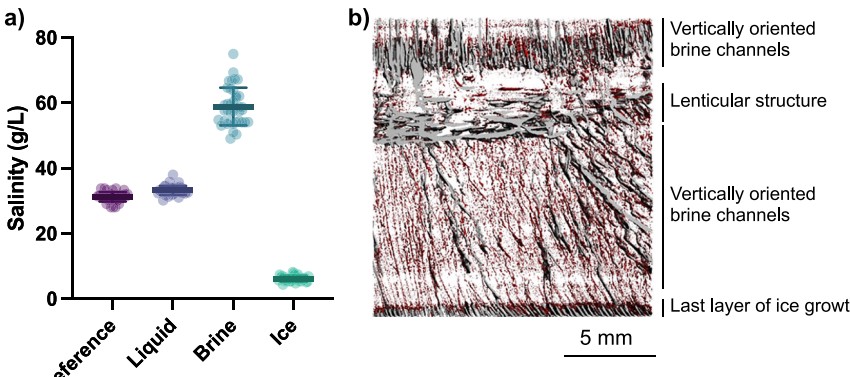

**Fig. 3 | Salinity and structure of the artficial ice cores. a** Salinity of the artificial seawater before freezing (Reference) and in the different compartments (Liquid, Brine and Ice) measured after freezing. There was significant evidence that the four salinities were different ($p < 0.05$). The horizontal line represents the mean and error bars represent the standard deviation. All data points across mulitple experimental replicates are shown, $n = 41$. **b** Example structure of a centrifugated ice core. Image was obtained by µ-CT and shows the ice (transparent), large porosity (gray) and small porosity containing entrapped brine (red).

artificial sea ice produced was very similar to young columnar sea ice in terms of bulk salinity, brine salinity, porosity and structure.

At the end of the experiment, the underlying liquid was the dominant fraction accounting for $93.09 \pm 1.27$ % of the total mass while ice and brine comprised $5.60 \pm 1.02$% and $1.04 \pm 0.34$% of the total mass, respectively (Fig. S2). This ensured that the salts which were expelled from the ice were strongly diluted in the underlying liquid, allowing ice to grow in realistic water salinities (Fig. 3, panel a). The brine salinity approximately doubled relative to the initial salinity, reaching $58.92 \pm 5.75$ g L$^{-1}$. This brine salinity was compared to predictions of brine salinity as a function of temperature[29]. We used $-1.7$ °C as the temperature at the bottom of the ice core since it is the freezing temperature of our artificial seawater ($31.28$ g L$^{-1}$) and $-5.1$ °C, as the temperature at the top of the ice core, since it is the temperature of the cooling element. This suggests that the brine salinity should be $57.80$ g L$^{-1}$ which is consistent with our observations. Conversely, the drained ice was strongly desalinated to $6.18 \pm 0.95$ g L$^{-1}$ (Fig. 3, panel a). The porosity of the sea ice, which corresponds to the brine volume fraction, was $0.15 \pm 0.04$. The sea ice bulk salinity (calculated using brine and ice salinity and volumes) was $13.86 \pm 2.30$ g L$^{-1}$. Note that since here "brine" refers to brine that is drained by centrifugation and "ice" refers to the residual drained ice, some small brine pockets or thin brine channels which were not effectively drained remained associated to and measured with the ice fraction.

The porosity and salinities of our artificial sea ice were very similar to those of natural young sea ice[39]. For example, in the Fram Strait, young sea ice had a median brine volume fraction of $0.12$ and a brine salinity of $80$ g L$^{-1}$ [40]. Furthermore, in a broad assessment of young lab-grown and field-grown sea ice, brine volume fractions varied between $0.07$ and $0.23$ and bulk salinities ranged from $10$ to $20$ g L$^{-1}$ [41]. Considering that the ambient temperature was $1$ °C and that the temperature of the top of the artificial sea ice was $-5.12 \pm 0.08$ °C, its bulk salinity of $13.86 \pm 2.30$ g L$^{-1}$, brine volume fraction of $0.15 \pm 0.04$ and brine salinity of $58.92 \pm 5.75$ g L$^{-1}$ fall within expected values of natural ice grown under similar temperature forcings[39,42]. For example, Notz and Worster[43] observed that sea ice of comparable thickness had a bulk salinity ranging from $15$ to $25$ g L$^{-1}$ [44]. Our lower bulk salinity may be explained by the fact that brine may be lost when the artificial sea ice is pulled out of the column. The solid mass fraction of their sea ice ranged from $0.5$ to $0.7$[44] which is lower than our solid mass fraction of $0.85 \pm 0.04$. This can again be attributed to some brine loss, as well as some residual brine pockets within the ice that were not connected to the bottom of the ice, and therefore not drained by centrifugation. Despite these experimental limitations, our results agree well with the

properties of natural sea ice. It should also be noted that this artificial sea ice produced represents the very early stages of sea ice growth, since young sea ice can reach up to $30$ cm depth and can desalinate to $4$ to $6$ g L$^{-1}$ [39,44]. Finally, the pH of the artificial sea ice cannot be directly compared to natural samples since biotic processes such as photosynthesis will modify pH, however, it is of interest to note that the changes in pH are not directly proportional to salinity (Fig. S3).

The ice cores had similar morphology to young columnar ice, despite variability between each ice core. The 3D rendition of the ice structure shows ice, air (where brine has been drained out) and some thin channels of brine which have not been drained by centrifugation (due to close-off of channels) (Figs. 3 panel b and S4). The top of each ice core had a first layer of short, vertically oriented brine pockets (-1 mm from the cooling element) followed by brine channels which are typical of those observed in natural sea ice[41,43]. The large channels (in gray) had mean widths of $121.2 \pm 30.4$ µm, while the thinner channels (in red) had mean widths of $50.5 \pm 14.1$ µm (Table S2). Some ice cores had a section of horizontally oriented and slightly curved lenticular structures which are not present in natural sea ice. This is likely caused by the fact that the ice started to grow on the freezing element which had a curved shape. This curved shape was used to allow air bubbles to escape as the cooling element was plunged into the water, to enable full contact between cooling the element and the water. Finally, some ice cores clearly showed the last of the ice growth with the parallel alignment of brine channels at the bottom.

## Model contaminants had different distributions in the sea ice column after freezing

The various contaminants differentiated strongly in their tendency to be incorporated in the sea ice, depending on whether they were dissolved (<1 nm), colloidal (1 nm to 1 µm) or particulate (>1 µm) (see Table S1. and Fig. S1 for images of model colloidal and particulate contaminants). First, the incorporation of each contaminant was assessed by comparing their concentration (C) in the bulk ice ($C_{bulk\ ice}$ which corresponds to brine and ice) normalized by their initial concentration in the column before freezing ($C_0$) (Fig. 4). As mentioned above, the ice bulk salinity was reduced to $0.44 \pm 0.07$ of the initial salinity. Most other contaminants are depleted from the bulk ice, albeit in varying concentrations. The model dissolved contaminant, Rose Bengal, and the model colloidal contaminant, NP, were depleted in statistically similar proportions as sea salts (with $C_{bulk\ ice}/C_0 = 0.43 \pm 0.01$ and $0.38 \pm 0.09$, respectively). This was followed by the other colloidal contaminant, nanosoot, which was depleted to $0.48 \pm 0.01$ of its initial concentration.

The fate of particulate contaminants, was assessed by comparing high-density MPs and low-density MPs of two different size-classes resulting in four different MPs. High-density MPs were composed of polyethylene terephthalate (PET-MPs; 1.39 g cm$^{-3}$) and low-density MPs composed of polypropylene (PP-MPs; 0.91 g cm$^{-3}$). All MPs were fragments produced by cryo-milling and sieving, resulting in area equivalent diameter of 35.6 ± 14.4 μm and 45.7 ± 20.7 μm for small PET-MPs and small PP-MPs, and 90.0 ± 29.8 μm and 72.3 ± 37.2 μm for large PET-MPs and large PP-MPs, respectively. When inserting PP-MPs, many particles remained at air-liquid interface due to their light weight and hydrophobicity. These PP-MPs did not interact with the advancing ice front as would be the case during bottom freezing in the environment. Therefore, this fraction of PP-MPs was removed from the evaluation of distribution in the column (Figs. 4, 5 and 6) but still considered when establishing the MPs recovery rate across the experiment (cf: Methods). When considering the portion of PP-MPs in the top layer of bulk ice, strong enrichment is observed as shown in Supplementary Fig. 5 ($C_{bulk\ ice}/C_0$ = 7.37 ± 2.53 and 3.22 ± 1.31 for Large and Small PP-MPs, respectively).

Overall, clear density-dependent trends are observed, even when excluding the top layer of ice with PP-MPs that could have been adsorbed to the air/water interface (Fig. 4) The high-density PET-MPs are depleted and the low-density PP-MPs concentrations remain almost constant. Levels of depletion/enrichment increase with particle size (Supplementary Section 1). The Large PET-MPs were significantly more depleted from the bulk ice than Small PET-MPs with $C_{bulk\ ice}/C_0$ = 0.06 ± 0.05 and 0.49 ± 0.25, respectively. This is explained by the fact that particle's settling or rising velocity is directly proportional to the power of its size. Conversely, Large PP-MPs had a higher $C_{bulk\ ice}/C_0$ value of 1.21 ± 0.59 than Small PP-MPs (0.86 ± 0.25).

To understand the mechanisms of contaminant incorporation in sea ice, we assessed the distribution of contaminants across the different sea ice fractions. To do so, brine and ice were separated to analyze the distribution of contaminants within the sea ice. The ratio of concentrations between brine and ice was measured as it reflects what happens at the freezing front and allows us to understand contaminants' incorporation mechanisms. Also, the brine fraction of sea ice is where sympagic organisms thrive due to the availability of nutrients, the space and (sometimes) incoming sunlight[22]. Consequently, the partitioning of contaminants between brine and ice are an important consideration when assessing exposure. Therefore, we first describe the distribution patterns of solutes, followed by the distribution pattern of particles since they contrasted most. This is followed by a discussion of the distribution of colloidal contaminants, since their distribution was in between those of solutes and particles (Fig. 5).

The model dissolved contaminant, Rose Bengal, followed the same distribution pattern between ice, liquid and brine as sea salts. That is, relative to the initial concentration, it was strongly depleted in ice ($C/C_0$ = 0.200 ± 0.012), enriched in the brine ($C/C_0$ = 1.528 ± 0.033) and only slightly enriched in the underlying water ($C/C_0$ = 1.009 ± 0.004) (Fig. 5). Rose Bengal and sea salt concentrations were not significantly different from each other in either ice or liquid ($p$ = 0.971 and 0.012, respectively). However, Rose Bengal was less enriched in the brine compared to salts which had a $C/C_0$ = 1.881 ± 0.210 ($p$ = <0.001). Note that since brine comprised only ~1% of the total volume, and since the mass balance was constant, even large changes in concentration in the brine fraction only resulted in small concentration changes in the ice and liquid. This similarity in behavior is consistent with prior findings showing that natural or anthropogenic dissolved species (e.g., salts, nutrients and tracer molecules) partition out of the ice and are enriched in the brine and

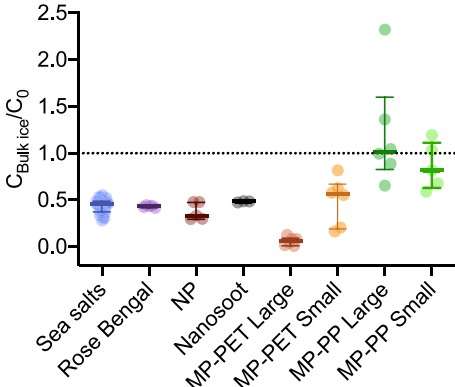

**Fig. 4 | Concentration of different species measured in bulk ice ($C_{bulk\ ice}$) relative to the initial concentration in the entire column ($C_0$).** Data points, median and IQR. $n$ = 41 for Sea salts, $n$ = 9 for PET-MPs Large, $n$ = 6 for NP, PET-MPs Small, PP-MPs Large and Small, $n$ = 5 for Rose Bengal, $n$ = 3 for Nanosoot.

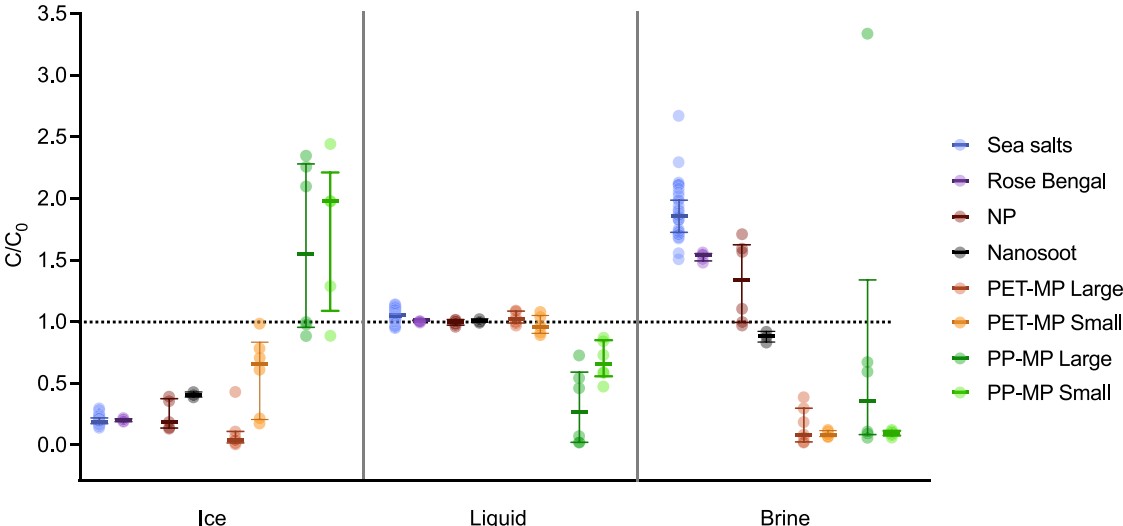

**Fig. 5 | Concentration of different species measured in each fraction (C) relative to the initial concentration in the entire column ($C_0$).** Data points, median and IQR. $n$ = 41 for Sea salts, $n$ = 9 for PET-MPs Large, $n$ = 6 for NP, PET-MPs Small, PP-MPs Large and Small, $n$ = 5 for Rose Bengal, $n$ = 3 for Nanosoot.

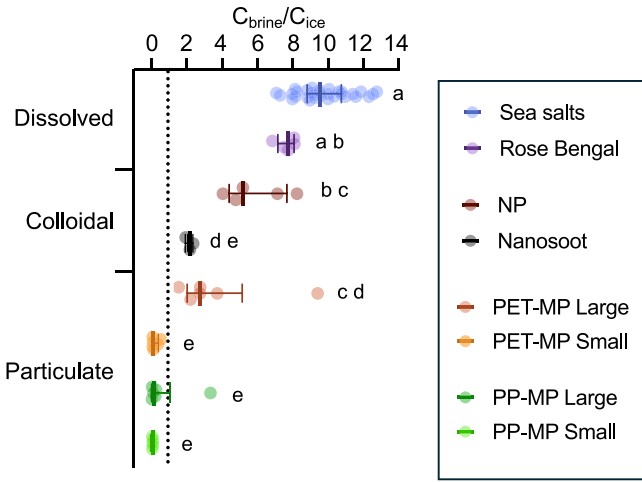

**Fig. 6 | Concentration of species in brine ($C_{brine}$), relative to their concentration in ice ($C_{ice}$) for each ice core.** Data points, median and interquartile range. Letters indicate statistical significance at the 0.05 level. $n = 41$ for Sea salts, $n = 9$ for PET-MPs Large, $n = 6$ for PET-MPs Small, PP-MPs Large and Small, $n = 5$ for Rose Bengal and NP, $n = 3$ for Nanosoot.

then depleted from bulk ice by gravity drainage[42,45]. Note that, as for the salts, it is also possible that some Rose Bengal in brine pockets was not centrifuged out of the ice. This would increase their measured concentration in the ice and decrease their measured concentration in the brine.

As demonstrated with the enrichment/depletion trends of MPs in bulk ice, MP's enrichment/depletion patterns are attributed to i) polymer density, which confer a positive buoyancy (i.e., rising) for the low-density PP-MPs and a negative buoyancy (i.e., settling) for the high-density PET-MPs and to ii) size, which accelerates the enrichment/depletion of MPs (Figs. 4, and S5, and Supplementary Section 1). This is further demonstrated by comparing concentrations in ice and in the underlying liquid. Both small and large high-density PET-MPs were depleted in the ice, with the Large PET-MPs more depleted than the Small PET-MPs ($C/C_0 = 0.046 \pm 0.038$ and $0.580 \pm 0.322$, respectively) (Fig. 5). Conversely, both Large and Small low-density PP-MPs were enriched in the ice ($C/C_0 = 1.594 \pm 0.707$ and $1.714 \pm 0.620$, respectively). No significant differences were observed between Large and Small PP-MPs' concentrations in ice. This could be due to the removal of the top 0.5 cm ice section from the analysis of particle distribution (due to its affinity for the air water interface, cf.: Methods) which introduced a higher variability in PP-MPs concentrations. PP-MPs' enrichment was compensated by a depletion in the underlying liquid ($C/C_0 = 0.308 \pm 0.308$ and $0.682 \pm 0.158$ for large and small PP-MPs, respectively). For PET-MP, no significant changes in concentration in the liquid fraction were observed, relative to the initial concentration ($C/C_0 = 1.027 \pm 0.051$ and $0.976 \pm 0.073$, for Large and Small PET-MPs, respectively) since the MPs were depleted from a small volume of ice and transferred into a large volume of underlying water.

In addition to density, MPs' size also determined the degree of enrichment/depletion in the brine fraction. Surprisingly, the concentrations of MPs in brine were not correlated to their concentrations in ice. Based on the ice morphology, particle size and rising/settling velocities, two effects could be at play. First, the speed of advancement of the freezing front relative to MPs' settling/rising rate could determine whether MPs have a chance of being located in brine channels and how easily they are drained from the channels by centrifugation. Since PP-MPs rose quickly in the column (Supplementary Section 1), most particles likely entered the ice when it started to form. This would lead them to either be engulfed at the top of the ice and out of the

brine channels or to be located at the top of the brine channels, where they would need to travel a longer and/or more tortuous path to be drained out by centrifugation. Second, the size of the particles relative to the brine channel width could explain different MPs depletion/enrichment ratios in brine and ice. Indeed, both small MPs are more depleted in the brine than larger ones ($C/C_0 = 0.811 \pm 1.267$ and $0.095 \pm 0.023$ for Large and Small PP-MPs, respectively $0.167 \pm 0.151$ and $0.088 \pm 0.024$ for Large and Small PET-MPs, respectively). This suggests that large MPs are physically trapped in brine channels and unable to settle or rise through them, while small MPs, which are on average smaller than all brine channels, can still move through them.

The colloidal contaminants had a distribution in ice and brine which was in between those of dissolved contaminants and the high-density PET-MPs. The extent of NPs depletion from ice ($C/C_0 = 0.242 \pm 0.124$) was not significantly different from salts ($p = 0.0098$) and was in between the values for Small and Large PET-MPs (Fig. 5). Nanosoot depletion from ice ($C/C_0 = 0.407 \pm 0.021$) was significantly less than for salts ($p < 0.001$) and, again, in between the values for Small and Large PET-MPs. In brine, both NPs and nanosoot were more enriched than any MPs but less enriched compared to salts ($p = 0.009$ and $p < 0.001$, for NPs and nanosoot relative to salts, respectively). Indeed, NPs were barely enriched with a $C/C_0$ of $1.324 \pm 0.334$, while nanosoot was even slightly depleted, with a $C/C_0$ of $0.876 \pm 0.044$. Finally, concentrations of NPs and nanosoot were relatively stable in the underlying water, again due to the same dilution effects observed for salts and PET-MPs ($C/C_0$ of $0.994 \pm 0.023$ for NPs and $1.007 \pm 0.015$ for nanosoot).

### Colloidal species' distribution depends on their aggregation state

The behavior of colloids in sea ice is highly dynamic since freezing can promote colloidal aggregation, as has already been already observed in batch experiments[46]. The distribution of colloidal contaminants, which is between those of dissolved species and high-density particulate species, can be explained by the co-occurrence of dispersed and aggregated colloids in the artificial sea ice column. On one hand, Brownian motion allows dispersed colloids to diffuse away from the freezing front, and therefore, undergo freeze rejection[47,48]. On the other hand, colloids aggregate in seawater and these aggregates can be large enough for their transport to be dictated by the Stokes regime rather than Brownian motion[49]. Indeed, after 19 h aggregates had formed as evidenced by the 5-fold increase in the hydrodynamic diameters of NPs and nanosoot in all fractions, relative to their initial size (Fig. S6). In the case of NPs, the aggregate diameter was directly proportional to the salinity of the dispersion media, whereas nanosoot aggregates were largest in the liquid fraction, followed by brine and ice. Despite the presence of these aggregates, it is anticipated that many individual, dispersed NPs and nanosoot remained in suspension since the size distributions of colloidal suspensions are overestimated by dynamic light scattering measurements when aggregates are present[50]. Microscopy analysis of thawed ice cores showed that nanosoot aggregates were significantly larger than NP aggregates (Fig. S7). This can explain the fact that the distribution of nanosoot between the ice, liquid and brine is more similar to that of high-density MPs than dissolved species (Fig. 6). Furthermore, the larger size of nanosoot aggregates could have led to their settling before dynamic light scattering measurements, which explains the smaller hydrodynamic diameters of nanosoot measured in the ice fraction (Fig. S6).

### Species' distribution between brine and ice is indicative of their incorporation process

To understand the incorporation mechanisms of contaminants, it is useful to specifically consider what happens at the freezing front by

comparing the relative concentration of each species in brine and ice for each the replicate experiments separately (Fig. 6). A pattern clearly appears, showing that sea salts are most accumulated in the brine, followed by Rose Bengal and NPs. This highlights that dissolved species and the dispersed colloids are expelled from the ice and accumulated in brine. It also suggests that the decreasing enrichment in brine could be related to the species' decrease in hydrodynamic diffusion coefficient as molecular weight increases (salt < Rose Bengal < NPs). Such trends have also been observed for per- and polyfluoroalkyl substances (PFAS), that is, contaminants composed of fluorinated carbon chains of variable lengths. Indeed, short-chain PFAS were more enriched in the brine relative to the solid ice matrix (corresponding here to a high $C_{brine}/C_{ice}$), while long-chain PFAS were enriched in the ice rather than the brine (corresponding here to a low $C_{brine}/C_{ice}$)[33]. In fact, when modeling these results, there appeared to be a stronger decoupling between concentrations of PFAS and salts in bulk ice as the molecular weight of PFAS increased[51]. This decoupling was strongly correlated to brine salinity, which suggests that PFAS could also be undergoing salting out, thereby reducing its diffusion coefficient[52]. Finally, the $C_{brine}/C_{ice}$ of MPs clearly showed that smaller MPs were more depleted from brine, presumably because they traveled through it without being physically trapped.

### Suitability of lab-scale experiments to study incorporation of contaminants in sea ice

This lab-scale experiment allowed us to generate environmentally representative artificial sea ice by basal growth and to assess the incorporation process of a variety of model contaminants initially dispersed in water. Inevitably, laboratory studies simplify complex environmental processes. To assess how these results translate to the environment, the following section discusses the strengths and limitations of our experimental design.

With respect to the type of ice grown, the advantage of our experimental design is that, by being relatively small, it allows many replicates to be performed since the freezing time and the amount of model contaminants required are reduced. However, this smaller size does not enable the generation of underwater currents, which occur naturally under sea ice. Despite this, the artificial sea ice formed is similar to natural ice. One must also keep in mind that this experimental set-up generated columnar ice, which grows below the layer of frazil ice and, as such, does not interact with contaminants at the air/water interface. However, this columnar ice is most common in the Arctic, which receives more anthropogenic contaminants than the Antarctic.

With respect to the type of observations that can be made, this design does not enable the use of methods that avoid the loss of brine as the sea ice is removed (i.e., by inserting a box under the sea ice as it is pulled out[53]). Furthermore, the use of centrifugation may not drain the brine pockets that are not connected to the bottom of the ice core. Therefore, the brine content is underestimated and the ice salinity is overestimated, compared to methods which measure brine content in-situ[44]. Future experiments could incorporate in-situ temperature and salinity measurements to record the ice growth and brine content in real time. However, since microorganisms also thrive in the brine channels that are connected to the underlying water, the brine fraction studied here is the more biologically-active fraction of the sea ice. Finally, this experimental design does not enable in-situ measurement of the adsorption of contaminants onto the ice or of the aggregation state of colloids and particles. Therefore, processes such as particle and colloid adsorption by hydrophobic or van der Waals attraction, or entrapment of aggregates in the brine channels cannot be observed. Future experimental developments that enable in-situ quantification and observation of colloids and particles in ice would allow to better quantify them and assess their surface affinity for ice and their aggregation state in ice.

### Environmental implications

Microalgae living in sea ice may be exposed to high loads of contaminants, adding to other stressors they must contend with (e.g., acidification and warming). Furthermore, the dynamics of contaminant incorporation into and transport with sea ice are changing with global warming, as young sea ice replaces multi-year ice. Therefore, we studied how contaminants present in seawater are incorporated in sea ice to assess exposure levels and understand new patterns of contaminant cycling. Specifically, this study investigated the processes occuring at the seawater/ice interface, which is most biologically active. Our study helps to predict a contaminant's incoporation into sea ice by taking into consideration its size, density (in the case of particulate species), and aggregation state (in the case of colloids).

Our results suggest that organisms which inhabit the porosity of sea ice will be highly exposed to dissolved contaminants and dispersed colloidal contaminants. While low-density micrometric contaminants were enriched in ice overall, they may only be temporarily bioavailable since they can travel upward through brine channels, and therefore be incorporated at the top of the ice, which is less porous due to brine drainage. Since the properties of sea ice, such as the speed of advancement of the freezing front and the ice porosity, play important roles in contaminant incorporation, measurements of contaminant concentrations in natural sea ice should be accompanied with a description of the sea ice structure and history to understand their transport routes. This can help to understand whether contaminant incorporation occurs during basal ice growth or frazil ice growth.

The processes by which contaminants' size and density drives their incorporation in sea ice, revealed in this study, are also applicable to natural species. This can help resolve long-standing questions about the incorporation of essential elements such as iron. Indeed, sea ice is an ideal environment for the aggregation of colloids and potentially also for the self-assembly of dissolved species into colloids or particles due to i) the local increase in salt and colloid concentrations in brine, and to ii) the growth of solid ice which can push colloids closer together thereby increasing the possibility of interaction. Therefore, the decoupling between "dissolved" (<0.4 μm) and "particulate" (>0.4 μm) iron observed in sea ice[54], as well as the higher enrichment of organic matter < 0.7 μm in brine relative to salts[55], could be due to the overlooked assembly of macro-molecules and aggregation of colloidal species. In conclusion, the results and methods presented here not only help to understand the fate of contaminants, but more generally sea ice biogeochemistry in the context of climate change.

## Methods
### Model contaminants

Rose Bengal Sodium salt (CAS: 632-69-9, Sigma-Aldrich) was used as the model dissolved molecule, with an initial conentration of $6.7 \pm 0.3\,\mu mol\,L^{-1}$ in the water column. A model carbon-based contaminant, a soot nanoparticle grafted with carboxylic functional groups (hereafter called nanosoot), was used with an initial concentration of 32 mg L$^{-1}$ (Aqua-Black 162® Tokai Carbon, Tokyo, Japan). The initial hydrodynamic diameter of the nanosoot was $117 \pm 3$ nm as determined by dynamic light scattering (DLS; Malvern Zeta-sizer). The zeta-potential was measured to be $-55.26 \pm 1.35$ mV, as determined by electrophoretic mobility (Malvern Zeta-sizer).

Model NPs and MPs were composed of polymers that were homogenously enriched with trace-metals, and were quantified as a proxy for the plastic using inductively coupled plasma mass spectrometry (ICP-MS). This approach allowed for quantification of trace concentrations of plastics more easily while also avoiding risks of NPs and MPs contamination within our experimental system. The low metal incorporation did not significantly change the particle density, and there was no leaching of metal over time (as shown in previous studies for NPs[56–59] and PET-MPs[59], as well as in Supplementary Section 2 for PP-MPs). NPs were composed of a polyacrylonitrile core and

a polystyrene (PS) shell with a final dopant concentration of $0.295 \pm 0.009$ w/w % palladium (Pd). The initial hydrodynamic diameter and zeta-potential of the NPs were $210 \pm 19$ nm and $-52.83 \pm 2.23$ mV, respectively. High-density ($1.39$ g cm$^{-3}$) MPs were composed of polyethylene terephthalate (PET) with 0.18 w/w % Indium (In) and low-density ($0.91$ g cm$^{-3}$) MPs were made of polypropylene (PP) with 0.14 w/w % Yttrium (Y). MPs fragments of both PET and PP were produced by cryo-milling and sieving at different mesh sizes. Large MPs were retained between 63 and 125 μm sieves, and Small MPs passed through a 63 μm sieve. The mean area equivalent diameter of Small PET-MPs and PP-MPs were $35.6 \pm 14.4$ μm and $45.7 \pm 20.7$ μm, respectively. Large PET-MPs and PP-MPs had a mean area equivalent diameter of $90.0 \pm 29.8$ μm and $72.3 \pm 37.2$ μm, respectively.

### Artificial sea ice production and processing

Artificial seawater (ASW) with a salinity of $31.28 \pm 1.47$ g L$^{-1}$ and pH of $7.96 \pm 0.16$ was prepared following the method of Kester et al.[60]. Natural organic matter composed of humic acid salt (solid, Acros Organic, CAS: 1415-93-6) and sodium alginate (solid, Sigma Aldrich, CAS: 9005-38-3) was prepared using the methods described in Pradel et al.[61] and added to ASW such that final concentrations were 0.7 mg L$^{-1}$ and 7 mg L$^{-1}$ respectively, to mimic conditions below sea ice and to improve the dispersion of particles and colloids.

Artificial sea ice was generated by a temperature gradient in a column under gentle agitation for 19 h (Fig. 2). The cylindrical columns were composed of polymethyl pentene with a diameter of 6.55 cm and height of 44.0 cm. After being filled with $1142 \pm 2$ mL ASW, an aluminum cooling element of 5.9 cm diameter and 4 cm depth was inserted onto the top of the column, in full contact with the water surface, with a slightly rounded shape to allow air bubbles to escape. The cooling elements, cooled by circulation of cooling liquid from a cryothermostat bath, had a bottom of the temperature of $-5.1 \pm 0.1$ °C. The experiments were conducted at an ambient temperature of $1.0 \pm 0.6$ °C, resulting in a temperature gradient over the column height (Fig. 2, panel a). The bottom of the sea ice column was fitted with a flexible PVC tube with an inner diameter of 3 mm which extended above the level of the column. This allowed the liquid to move up the tube as the ASW froze and expanded, thereby avoiding pressure increases in the system. The columns were set on an orbital shaker (KS 15 A, Huber Lab) at 50 rpm. For each experimental condition, three columns with cooling elements were aligned in series (Fig. 2, panel b).

Freezing lasted 19 h to ensure that the ice thickness attained a plateau which was typical for sea ice grown under constant temperature forcings (Fig. S8). At the end of the experiment, the ice had a mean depth of $3.32 \pm 0.61$ cm (Fig. 2, panel c). The ambient temperature of the experimental room was lowered to $-2.5$ °C and the bulk ice was lifted out of the column with a solid net attached to two nylon strings. Artificial sea ice cores were immediately centrifuged (Sigma 3-16KL) for 10 min at 40 g at $-3.5$ °C to drain the brine connected to the bottom of the bulk ice core. This temperature was selected since based on the ASW concentration and the temperature gradient, the brine salinity was estimated to be ~60 g L$^{-1}$ (cf: Figure 12.1 of ref. 40) and 60 g L$^{-1}$ ASW freezes at $-3.5$ °C[62,63]. Therefore, centrifugating above the freezing point of the brine and below the freezing point of pure ice, allows us to preserve the ice core structure. However, sea ice salinity and permeability is complex and nonlinear[64], and there is inherent experimental variability in the temperatures. For example, measured temperatures were $-2.3$ °C at the top and $-4.6$ °C at the bottom in the centrifugation bucket. Therefore, strictly speaking, the brine obtained represents the brine with freezing points above the actual centrifugation temperature. The mass of centrifugated brine, centrifugated ice and underlying liquid were weighed after centrifugation.

To characterize the structure of the sea ice, several centrifugated ice cores were set aside at $-20$ °C and scanned by micro-X-ray-tomography (μCT)[43]. By draining the brine out of the ice prior to scanning, the porosity of the ice was more visible since the contrast between air and ice is higher than between liquid and ice. However, some brine was distinguishable from the ice. Therefore, a 3D rendition of the sea ice structure was created by segmenting into three different levels of absorption to observe ice, drained brine channels, and residual brine. μCT scans were performed on a subsample of 3 cm diameter of the ice core using a μCT 40 by SCANCO Medical AG (Brüttisellen, Switzerland). X-ray tube settings of 70 kVp and 8 W were used. For all samples, a total exposure time of 600 ms was used with 1000 projections per slice (180° of rotation). The resolution was 11 μm. The images were reconstructed using a proprietary back-projection algorithm. The 16-bit images were filtered by a Gaussian 3D-filter with a size of 4^3 pixels and a standard deviation of 1.2 pixel. Segmentation of the air-ice threshold was 22 permille, of the ice-brine threshold 72 permille. Further visualization was done in ImageJ[65].

### Quantification of contaminant distribution in artificial sea ice

To evaluate species' rates of incorporation into sea ice, we performed experiments with one species at a time and assessed their concentrations in thawed ice, liquid and brine at the end of the experiment relative to their initial concentration in the column ($C/C_0$). To quantify the distribution of species at the freezing front, we also assessed the relative concentrations in brine and ice ($C_{brine}/C_{ice}$) in each column separately. Given the physicochemical differences of the model species, different methods were used to initially disperse, recover and quantify them.

Dissolved and colloidal species (Rose Bengal, NPs, and nanosoot) were dispersed into a master batch (i.e., a large volume homogenized by mixing) which was then equally distributed between the three columns. Particulate species were either positively buoyant (PP-MPs) or negatively buoyant (PET-MPs) and consequently could not be homogenously distributed within the columns. To ensure that sufficient quantities of each MPs type would be interacting with the freezing front we either added PP-MPs to the bottom of the column with 100 mL of ASW before completing with the rest of the ASW volume, or we added the ASW and then added the PET-MPs to the top of the column with the last 100 mL. When inserting PP-MPs into the liquid prior to freezing, many PP-MPs remained at air-liquid interface due to their light weight and hydrophobicity. This resulted in PP-MPs at the top of the sea ice cores which did not interact with the advancing ice front. Therefore, the top 0.5 cm of the ice core was removed after centrifugation and excluded from the evaluation PP-MPs distribution in sea ice but still considered when establishing the PP-MPs recovery rate across the experiment.

Species were recovered and their concentrations measured in each fraction (i.e., brine, ice, liquid) using different analytical approaches. Seawater salts were quantified by measuring electrical conductivity (FiveEasy conductivity meter, Mettler Toledo) and converting electrical conductivity to salinity, using a calibration curve. Rose Bengal and nanosoot were quantified by absorbance spectroscopy (at $\lambda = 548$ nm and $\lambda = 400$–600 nm, respectively, Cary 60 UV–vis, Agilent). The analytical workflow for plastic contaminants consisted of concentration, digestion, and quantification of their metal content. MPs were recovered on filters (Macherey-Nagel 615, with a cut-off of 4 to 12 μm). NPs were pre-concentrated by flocculation using poly(acrylamide co-acrylic acid) (Sigma-Aldrich) and aluminum nitrate nonahydrate (Sigma-Aldrich). The protocol had a recovery rate of $95 \pm 5\%$, and is further described in Supplementary Section 3. Filters with MPs and NP flocs underwent microwave acid digestion in an ultraWAVE digestor (Milestone Srl, Italy) with either 2.3 mL or 5 mL distilled HNO$_3$ 65% (puriss. Sigma-Aldrich), respectively. The temperature and pressure increased from ambient to 220 °C and 160 bar over 30 min and were then maintained at 220 °C and 160 for 30 min. Digestates were then diluted and quantified by ICP-MS (Agilent 7900). For each element measured

($^{106}$Pd, $^{115}$In and $^{89}$Y) a calibration curve and quality controls were performed daily. Internal standards of Scandium, Yttrium and Rhodium (10 μg/L) were added to the Pd and In containing samples, while only Sc and Rh (10 μg/L) were used for Y-containing samples. Internal standards were continuously monitored throughout the measurements and results were corrected to account for instrumental drift.

The mass balance for most species was high (ranging from 95.75 % to 101.97 %), which gave us confidence that the distributions between the different fractions could be used to track the fate of the contaminants in the sea ice column (Table S3). For large PET-MPs, small PP-MPs and large PP-MPs the recoveries were lower (73.60 to 89.86 %). For PP-MPs, this can be explained by their hydrophobicity, which caused them to adsorb onto surfaces of the column and the air/water interface. The addition of model contaminants did not significantly affect the mass, pH, salinity or structure of the ice (Table S4 and Figs. S9 and S10).

### Assessment of the aggregation state of colloids

Following freezing, the aggregation state of the colloids (nanosoot and NPs) was assessed by measuring the size by DLS and zeta-potential by electrophoretic measurement using the Smoluchowski model (Malvern Zeta-sizer). The larger aggregates were measured by light microscopy using a ZEISS Axiolab microscope at a 10x magnification, an CamErc5s camera and the ZEN lite software. Long exposure times (300 to 1000 ms) were used to increase the visibility of small aggregates. The aggregates were characterized by analyzing the images with the NanoDefine particle sizer plugin[66,67] of the ImageJ software[65].

### Data analysis

Data was plotted either using R Statistical Software[68] or Prism software. To compare the concentrations of model contaminants in different fractions, two approaches were employed. First a F-Test for equality of two variances was performed. If variances were unequal at the 0.05 significance level, then a t-tests was performed for unequal variances. Otherwise a t-test with equal variances was performed. To quantify the speed of advancement of the ice front, a power equation was fitted by linear regression to the data.

## Data availability

The mass, pH, concentrations of salts and model contaminants in each fraction (bulk ice, ice, liquid, brine), as well the recovery rates data generated in this study are provided in the Supplementary Information/Source Data file[69] https://doi.org/10.6084/m9.figshare.27834345.

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

## Acknowledgements

AP was funded by the ETH Postdoctoral Fellowship and the Rütli Foundation. DMM was funded through the Swiss National Science Foundation (SNSF), grant number PCEFP2_186856. The authors gratefully acknowledge Michael Rösch for his help designing the artificial sea ice columns, Jana Rüthers and Sebastian Dotterl for the use of the climate chamber, Björn Studer for his help with ICP-MS analysis, Sithiprumnea Dul and Benno Wüst for MPs preparation, Evgenii Salganik for his explanation of sea ice growth. Thanks to Anne Greet Bittermann at the ScopeM platform for the microscopy images.

## Author contributions

A.P.: Conceptualization, Formal Analysis, Funding Acquisition, Investigation, Methodology, Validation, Visualization, Writing (Original Draft Preparation), Writing (Review and Editing); R.H.: Resources, Investigation, Writing (Review and Editing) M.S.: Resources, Investigation, Writing (Review and Editing); D.M.M.: Conceptualization, Project Administration, Funding Acquisition, Writing (Review and Editing).

## Competing interests

The authors declare no competing interests.
