## [Transparent Peer Review file · Nature Communications]

Impact of contaminant size and density on their incorporation into sea ice

Corresponding Author: Dr Alice Pradel

Version 0:

Reviewer comments:

Reviewer #1

(Remarks to the Author)

The study entitled "Understanding Contaminant Incorporation in Sea Ice: Impact of Size and Density" investigates the ice-water distribution patterns of model contaminants during freezing processes. The approach of separating ice and brine based on different freezing points demonstrates a certain degree of innovation. The microtomography has been used to characterize the structure of ice and the distribution of brine channels within it. The paper specifically discusses the influence of size and density on the distribution characteristics of model organic compounds in ice, brine, and water. However, several issues need to be addressed to improve the manuscript. It is recommended to revise and improve the paper from the following aspects:

1. In lines 389-390, the ice core was extracted using a solid net. The solid net is made of plastic material. Could it release or adsorb microplastics and nanoplastics, thereby affecting the results?
2. Is this ice core the entirety of the ice that had been frozen for 19 hours? Additionally, a detailed explanation is required regarding how the solid net separated the ice from the water in the mixture of ice and water. During the ice-water separation process, will the ice melt? Do the materials used in the simulation experiment affect the quantitative results? There should be a control experiment.
3. When considering the separation of brine and pure ice, differentiating them based on different freezing points is an interesting concept; however, several issues arise. The selection of the freezing point is one such issue. The salinity of the brines in different brine channels may not be entirely uniform. Choosing -3.5°C as the freezing point merely indicates that ice with a freezing point above -3.5°C will melt and detach from the ice core. Therefore, strictly speaking, the concentration of the brine obtained represents the concentration of only some brine channels, specifically those with freezing points above -3.5°C . Furthermore, during the separation process within brine channels, microplastics or nanoplastics may not be completely expelled and may partially adhere to the interstitial spaces between ice crystals. Hence, the brine concentration discussed in this context can only be considered as the 'bulk brine' concentration rather than the concentration of all brine. Moreover, this aspect requires supporting citations from relevant literature to enhance credibility, such as references that justify the choice of the freezing point.
4. In the supplementary materials, the author clarifies the use of microtomography for analyzing and calculating the widths of brine channels, which ranged from 50 to 240 μm . The selected microplastics in the study exhibit varying particle sizes, potentially leading to differing final results. Some microplastics may not be effectively separated from the brine channels, and there may exist brine channels with even smaller widths. These factors have a significant impact on the separation of brine and ice, as well as the ultimate outcomes of the study.
5. What is the significance of separating brine from ice, considering that in nature, brine and ice exist as an integrated entity? The environmental implications of centrifuging brine are unclear, and the separated brine only represents a portion of the total.
6. Figure 4 is missing. The text and figures do not match.
7. Lines 230-232: Microplastics of small particle sizes may not all be able to pass through brine channels, and they may

adhere to the edges of brine channels and ice. Additionally, aggregation of small-sized microplastics due to unstable freezing, which leads to an increase in relative particle size, may also trap them within brine channels.

8. When discussing the occurrence concentrations of different types of microplastics, particle size and density are crucial factors to consider. Additionally, zeta potential and contact angle are also key characteristics for describing the properties of plastics. A more comprehensive discussion of the mechanisms should be included when analyzing the distribution of microplastics in ice, brine, and water.

9. In Figure 3, the data related to polypropylene (PP) exhibit significant errors. For instance, the data points in brine deviate substantially from the main data cluster, suggesting the possibility of man-made error rather than systematic error. It is advisable to repeat the experiments to eliminate the large man-made error.

Reviewer #2

(Remarks to the Author)

Review of: Understanding Contaminant Incorporation in Sea Ice: Impact of Size and Density, by Pradel et al.
Max Thomas

Pradel et al. present experimental results from a new sea-ice growth facility. A suite of contaminants with a range of physico-chemical properties were added to the experimental system, and their concentrations were measured in the underlying liquid, brine centrifuged from sea-ice samples, and in the sea ice remaining after centrifugation. Pradel et al. then interpret the concentration measurements based on the range of properties in the contaminants.

The experiments and measurements look competently done and are presented well. The choice of contaminants seems very well thought out and allows for a detailed and novel analysis of the mechanisms behind the behaviour of these chemicals. Sections 2.2 – 2.4 are particular highlights which have potential to be highly influential on future work. The work is original, and provides important, incremental knowledge gains for the sea-ice contaminant and process modelling community.

There is some work to be done on improving clarity, and to explicitly state how (unavoidable) biases and limitations in the measurements impact the conclusions. I was also confused by the lack of any bulk sea ice values being presented (the only major comment, which would require some more data analysis) which would help comparison with other work and might help simplify and strengthen conclusions.

I recommend the work for publication after the revisions listed below, which are fairly minor.

Major comments:

1) Line 22 – I don't think this statement is justified: 'While... particulate contaminants were enriched [or depleted] in sea ice'. Normally, 'sea ice' would here refer to the bulk value, which would be the concentration measured in melted ice and brine together. Bulk values aren't presented. I presume figure 3 is evidence to support this statement but I don't understand how the results link back to the statement, as a sea ice measurement is not presented. I suggest calculating 'sea ice' using the measured concentrations of a contaminant in ice and brine and the measured mass or volume of ice and brine. Possibly evidence is there already for this statement, and if so that needs to be made clear.

Minor comments:

- 1) Line 12 – Sea ice forms near human activity too (Canadian towns, Longyearbyen) so please remove 'despite being far from human activity'. You may want to add that it transports contaminants as this is important and mentioned prominently in the intro.
- 2) Line 14 – Please make clear what aspects are 'novel'. The suite of chemicals is novel, but the cold plate, growth in tubes, artificial salt solution are not novel. If it is some combination of the design choices that is novel for some specific novel need, that should be made clear and justified. The methods don't need to be novel for the work to be novel and worthwhile.
- 3) Line 18 – What precisely does 'difficult' mean? Are you referring to contamination issues?
- 4) Line 20 – Are kDa and μm related to each other somehow? I don't see how a range of values from x to y is useful when x and y have different units.
- 5) Line 58 – The authors will need to be more clear about what 'brine rejection' is referring to. Generally, 'brine rejection' refers to brine moving from the sea ice to the ocean below, which desalinates the sea ice. This happens after sea ice has grown and incorporated everything. There is insignificant segregation at the advancing interface, so everything that was in the water is incorporated into the growing sea ice, then rejected (Notz & Worster, 2009, <https://doi.org/10.1029/2008JC004885>). What the authors are describing is salt and other dissolved species being excluded from the ice matrix such that they are partitioned to the brine, but remain within the sea ice.
- 6) Line 59 – The authors should be careful to use 'ice' to mean the frozen, ice matrix of sea ice, and 'sea ice' or 'bulk sea ice' to mean the collection of ice and brine. Here, 'bulk ice' is used, which is confusing.
- 7) Line 62 – (see minor comment 4)
- 8) Line 77 – 'Particulate species with low densities are expected to be enriched...' Enriched relative to what? Where does this hypothesis come from? The other hypotheses follow from the introduction, but this one seems unconnected to the previous introduction material.
- 9) Line 83 – This is probably the place to address minor comment 2.

- 10) Line 90 – Defining ice and brine earlier could solve previous comments.
- 11) Line 92 – ‘Trapped’ probably isn’t the right word here.
- 12) Line 92 – ‘Collectively this’ is imprecise (and should be ‘these’ I think). I would suggest replacing this with ‘Our results’ or ‘These knowledge gains’, if that’s what is meant.
- 13) ‘4’ is repeated in the figure 1 caption and ‘2’ is missing. Also ‘column’ should be ‘a column’ or ‘columns’. This figure is very clear and informative.
- 14) Line 108 – ‘vertical temperature gradient’.
- 15) Line 114 – ‘average’ to ‘mean’ (I presume).
- 16) Line 129 – ‘brine salinity approximately doubled’. The authors should calculate and report to temperature in Vancoppenolle et al. (2019). This comparison will help the authors think about how representative their centrifuged brine is of the in situ brine.
- 17) Lines 133-135 – It is great that the authors have thought deeply about this aspect of the experiment and have noted it here. It then needs to feature in the discussion (see also point 2.4).
- 18) Lines 136-143 – These values are reassuring, and suggest the artificial sea ice is a reasonable approximation of the real world. I would note that the sea ice in this study is very young (<5 cm), whereas real world sea ice is considered ‘young’ while less than 30 cm – I would expect 30 cm sea ice to have lower salinity and solid fraction. Also, the values reported by the authors here will be biased by brine loss during sampling (so before centrifuging). This bias is practically unavoidable. These caveats should be noted here. The authors could compare to Notz & Worster (2008) which present in situ measurements of bulk salinity and solid fraction (<https://doi.org/10.1029/2007JC004333>), and have measurements in <5 cm sea ice. For comparable thicknesses the authors will see the in situ measurements will have a higher bulk salinity and solid fraction.
- 19) Line 155 – Why did the freezing element have a curved shape?
- 20) Figure 2b – Could the authors please annotate the image? I had to guess what the lenticular structures were. Also, what is ‘black’ in the image?
- 21) Line 171 – ‘if’ to ‘is’
- 22) Line 229 – Needs an ‘and’ between the reported values.
- 23) Line 255 – ‘already’ to ‘already been’
- 24) Section 2.4 – As a minimum, I would like to see a line here reminding the reader that Cice in this study includes brine in isolated pockets. It would also be very useful to get the authors take on how the inclusion of isolate brine within Cice impacts their conclusions. I found it difficult to understand how, for example, the exposure of microorganisms to contaminants is impacted by having this imperfect separation of phases in the measurement. The authors should also note bias from brine loss upon sampling and speculate as to impacts on conclusions. Possibly these caveats could form a short paragraph, as this paragraph is already quite complex and well put together.
- 25) Figure 5 is excellent.
- 26) Line 323 – ‘pressure of crystal growth’? Do the authors mean actual pressure (as in Pa), and if not, what?
- 27) Glossary: The statements about this sea ice containing no gas should be softened. Gas is often found in young sea ice, and there could be some here, unless the authors have some evidence to the contrary.
- 28) At the end of the paper I realised I don’t understand why the low density MP were so depleted in the liquid. Scanning back through I couldn’t find the answer. Could the authors please explain this, or make the explanation more prominent.
- 29) I would also suggest saying ‘: The impact of...’ in the title as it’s plainer English.
(Just a note to say I thought the section titles were very helpful. The declarative statements made the paper easy to follow.)

Reviewer #3

(Remarks to the Author)

Understanding Contaminant Incorporation in Sea Ice: Impact of Size and Density
Pradel et al

This manuscript details the experimental set-up and application of an artificial sea ice chamber system that allows the intimate study of solutes and particles, present in sea water, and their interaction with freshly forming ice. The authors are to be applauded for their relatively simple but elegant experimental set up. To my knowledge this small-scale system with multiple plastic polymer columns, an air-side cooling system (representing real Polar conditions over surface seawater) combined with a pragmatic pressure release tube is novel. Given the chemical and physical measurements of the fresh ice plug (with CT imaging) then similarities to initial ice in the Arctic is convincing. Hence this column system does provide a unique system to study the uptake, release and fate of both dissolved and particulate contaminants in ice (e.g. persistent organic chemicals, plastic particles, POC/DOC, heavy metals etc). The authors then demonstrate the utility of the apparatus by examining the fate of selected solutes, colloids and plastic particles (varying density) during a freeze experiment. Enrichment/depletion of these species relative to their occurrence in seawater is then reported with comparisons, where possible, to field and laboratory studies. The study is at the standard for publication however there are several issues that the authors need to comment on or address before publication could proceed.

1) The columns and their seawater contents were agitated by being placed on an orbital shaker during the freeze period. It is not entirely clear why this was undertaken but does make sense in that particulate matter is likely to remain suspended in the water column, rather than deposited to the vessel floor (for the higher density particles). This agitation however does not represent a sub-ice seawater current though. This could be achieved using a small submerged pump or similar, which is common in larger sea ice chambers, which hence do resemble the surface ocean with this regard. On this note there doesn’t appear to be incorporation of sensors to measure physical parameters of the seawater itself. Most notably the temperature/salinity profile down the water column should be essential. For example, the expulsion of brine with ice aging or the release of low salinity melt water during ice melt, may result in stratified water layers which will affect chemical/material

flows and yet are not easily distinguished in the current system.

2) The freezing process using the aluminium cooling vessel will interfere with air side transfer of particles and vapour/gaseous species. While the focus is clearly on studying the ice itself and the ice-seawater interface, this is nonetheless a significant oversight particularly for studying the transfer of material/chemicals across the ice-air interface. Importantly, the introduction of material to the surface of the ice to simulate ice-rafted snow for example cannot be undertaken with the current set up.

3) In the Introduction the authors cite those studies that have observed microplastics in Polar sea ice at concentrations orders of magnitude higher than concentrations present in the underlying sea water. This is one of the drivers behind experimental simulations of the seawater/sea ice system like this study. However, in the experimental work for the microplastics the authors discarded the uppermost 0.5cm ice layer on the basis that lower density particles had accumulated in the water-air interface and hence had become artificially rafted onto the ice-surface. Surely, this is the very process which will be occurring to some extent for positively-buoyant plastic particles in the real marine environment! Does this ice-rafting of some types of microplastics occur during early formation of grease/frazil ice?? This is a key question which hopefully this new experiment apparatus could investigate, but which the authors have viewed as an artifact!!

P1, line 28 contaminantation

P2, line 72in [a] controlled environment...

P2, line 94wil be exposed to.

Version 1:

Reviewer comments:

Reviewer #1

(Remarks to the Author)

As the authors have repeatedly stated, the simulation experiments indeed have many limitations, but they have made the greatest possible effort to address my comments. I recommend that this version of the manuscript can be published.

Reviewer #2

(Remarks to the Author)

The authors have thoroughly responded to my first review, and I am happy that the manuscript is fit for publication.

There are three further points I suggest they address:

1) Comment 2, new lines 94-97: There are existing systems which keep pressure from building, so this aspect is not novel (Thomas et al. (2020) for example). The chemical suite should be mentioned here as that is novel. The curved cooling plate and net are also new to the sea-ice literature I think.

2) Figures 4 and 5 should be combined as they share axes and this would aid comparing them.

3) On lines 24-25 in the abstract, where the enrichment/depletion is discussed, it should be made clear that this is relative to salt.

Reviewer #3

(Remarks to the Author)

I have now reviewed the authors' responses to the comments made on the manuscript and they have satisfactorily addressed the points that I raised and as well the points raised by the other reviewers. Importantly the revised manuscript reflects this with substantial additions/changes made to the Results & Discussion. These additions improve the manuscript markedly and now highlight the limitations and/or biases that occur in the experimental ice columns relative to the real environment. The rationale for excluding or removing the ice rafted plastic particles is now clearly articulated. Overall I recommend that this manuscript can now be published.

There are a couple of typos in the following lines...

377 "With respect to the type [OF] observations that can be made, this design does not enable the use

388 of methods that avoid the loss of brine as [it] the sea ice is removed"

Reviewer #1 (Remarks to the Author):

The study entitled " Understanding Contaminant Incorporation in Sea Ice: Impact of Size and Density" investigates the ice-water distribution patterns of model contaminants during freezing processes. The approach of separating ice and brine based on different freezing points demonstrates a certain degree of innovation. The microtomography has been used to characterize the structure of ice and the distribution of brine channels within it. The paper specifically discusses the influence of size and density on the distribution characteristics of model organic compounds in ice, brine, and water. However, several issues need to be addressed to improve the manuscript. It is recommended to revise and improve the paper from the following aspects:

We thank the reviewer for their time to make detailed comments and suggestions to improve our manuscript, which will aid future readers in understanding the rationale of our experimental design and important aspects relating to particle physiochemical dynamics.

1. In lines 389-390, the ice core was extracted using a solid net. The solid net is made of plastic material. Could it release or adsorb microplastics and nanoplastics, thereby affecting the results?

While the plastic net could potentially release microplastics (MPs) and nanoplastics (NPs), an important distinction is that we used model MP and NPs in this experiment which are doped with trace metals. Subsequently, by quantifying the trace metals as a proxy for these model particles potential contamination from non-target MP and NP can be excluded. This is also true for potential release of MPs and NPs from the column, which is also made of plastic.

The plastic net (and column) could adsorb MPs and NPs. However, the loss of particles throughout the experimental workflow was minimal, as discussed lines 568 to 573 and shown in Supplementary Table 3. For salts and most model contaminants (Rose bengal, NPs, Nanosoot and PET) the mass balances is very close to 100%. For Small-PP and Large-PP mass balances are lower and more variable : 73.60 +/- 20.38 % and 88.49 +/- 35.28 % respectively. We have observed that some of these particles are retained at the air/water interface and on the column and suspect that this is due to their hydrophobicity. Therefore, we suspect that most of the loss does not occur at the seawater/ice interface, and that it has minimal effect on our conclusions obtained from Figures 4, 5 and 6.

2. Is this ice core the entirety of the ice that had been frozen for 19 hours?

Yes, the entire ice core which has formed over 19 hours has been recovered. We have now clarified this point in the text, see line 125-126.

Additionally, a detailed explanation is required regarding how the solid net separated the ice from the water in the mixture of ice and water.

We added an explanation on how the ice was separated in both the Results and Materials and Methods sections.

Lines 125-126: The entire ice cores were lifted out of the column with a solid net attached to two strings and were centrifugated (Sigma 3-16KL) to drain the brine connected to the bottom of the ice core.

Lines 494 - 496: The ambient temperature of the experimental room was lowered to -2.5 °C and the ice was lifted out of the column with a solid net attached to two nylon strings.

During the ice-water separation process, will the ice melt?

It takes less than 2 minutes to pull the ice out of each column and put each ice core in a styrofoam cooler at -3.5 °C before centrifugation. Therefore, we do not expect the ice to melt to an extent which would alter our results under this time frame.

Do the materials used in the simulation experiment affect the quantitative results? There should be a control experiment.

We had conducted experiments without model contaminants to assess the variability of the ice produced. When comparing the mass, salinity and pH of ice and brine for contaminant-laden conditions, we did not observe systematic differences with the control.

Furthermore, to assess whether different contaminants generated different ice structures, micro-CT scans were made for different experimental conditions. These included nanosoot (Fig S4.a), nanoplastic (Fig S4. b), large PP (Fig S4.c) and large PET (Fig. 3b). As discussed in Lines 177-189, the ice cores displayed similar structural features. Collectively, this suggests that the contaminants do not modify the structure of the ice formed under our experimental conditions. This was expected considering the concentration of contaminants was on the order of mg/L while the concentration of artificial sea salts was on the order of g/L.

We have now added the controls (i.e., contaminant-free variants) to the supplementary data and added this to the discussion to the Materials and Methods section which presents the production of artificial sea ice.

Lines 573 – 574 : The addition of model contaminants did not significantly affect the mass, pH, salinity or structure of the ice (Table S4 and Fig. S9 and S10).

3. When considering the separation of brine and pure ice, differentiating them based on different freezing points is an interesting concept; however, several issues arise. The selection of the freezing point is one such issue. The salinity of the branches in different brine channels may not be entirely uniform. Choosing -3.5°C as the freezing point merely indicates that ice with a freezing point above -3.5°C will melt and detach from the ice core. Therefore, strictly speaking, the concentration of the brine obtained represents the concentration of only some brine channels, specifically those with freezing points above -3.5°C. Furthermore, during the separation process within brine channels, microplastics or nanoplastics may not be completely expelled and may partially adhere to the interstitial spaces between ice crystals. Hence, the brine concentration discussed in this context can only be considered as the 'bulk brine' concentration rather than the concentration of all brine. Moreover, this aspect requires supporting citations from relevant literature to enhance credibility, such as references that justify the choice of the freezing point.

We thank the reviewer for this comment, and indeed it was a point which we discussed internally when designing the experiments and interpreting the results. Here, the objective was to separate brine and ice to analyse the distribution of contaminants within the sea ice porosity. The separation of brine from the ice core was done at a temperature that minimized changes in the brine content of the artificial sea ice. Considering the salinity of our artificial seawater ($31.28 \pm 1.47 \text{ g L}^{-1}$) and the temperature of the freezing element ($-5.1 \pm 0.1 \text{ }^\circ\text{C}$), brine with a maximum salinity of 62 g L^{-1} was expected to be formed (cf: Figure 12.1 of Thomas, D. N. & Dieckmann, G. S. (2010)). We then calculated the freezing point of seawater at 62 g kg^{-1} , with a sea pressure p (defined as the Absolute Pressure P less the Absolute Pressure of one standard atmosphere) of 0, and an air saturation of 1, using the Gibbs Seawater (GSW) Oceanographic Toolbox. This results in freezing temperature of $-3.52 \text{ }^\circ\text{C}$. Therefore, we set the temperature of the centrifuge at the freezing temperature to ensure that the brine would not freeze and the ice would not thaw. Please note that this temperature is within the validity domain of the GSW equations, as explained in Vancoppenolle et al. (2019). We have now explained this reasoning, provided references and discussed limitations in the main text :

Lines 498 to 506: This temperature was selected since based on the ASW concentration and the temperature gradient, the brine salinity was estimated to be approximately 60 g L^{-1} (cf: Figure 12.1 of ref 39) and 60 g L^{-1} ASW freezes at $-3.5 \text{ }^\circ\text{C}$. Therefore, centrifugating above the freezing point of the brine and below the freezing point of pure ice, allows us to preserve the ice core structure. However, sea ice salinity and permeability is complex and nonlinear⁶⁵, and there is inherent experimental variability in the temperatures. For example, measured temperatures were $-2.3 \text{ }^\circ\text{C}$ at the top and $-4.6 \text{ }^\circ\text{C}$ at the bottom in the centrifugation bucket. Therefore, strictly speaking, the brine obtained represents the brine with freezing points above the actual centrifugation temperature.

We agree that that brine concentrations only represent concentrations in the brine that could be drained out and not concentrations in residual brine pockets. We have mentioned this experimental limitation throughout the manuscript:

Lines 165 – 167 : This can be attributed again to some brine loss, as well as some residual brine pockets within the ice that were not connected to the bottom of the ice, and therefore not drained by centrifugation.

Lines 261 – 264: Note that, as for the salts, it is also possible that some Rose Bengal in brine pockets was not centrifugated out of the ice. This would increase their measured concentration in the ice and decrease their measured concentration in the brine.

We have also discussed our experimental limitations in section 2.5 entitled “Suitability of lab-scale experiments to study incorporation of contaminants in sea ice”

Lines 387 – 392: With respect to the type observations that can be made, this design does not enable the use of methods that avoid the loss of brine as it the sea ice is removed (i.e.: by inserting a box under the sea ice as it is pulled out⁵⁴). Furthermore, the use of centrifugation may not drain the brine pockets that are not connected to the bottom of the ice core. Therefore, the brine content is underestimated and the ice salinity is overestimated, compared to methods which measure brine content in-situ⁴³.

4. In the supplementary materials, the author clarifies the use of microtomography for analyzing and calculating the widths of brine channels, which ranged from 50 to $240 \text{ }\mu\text{m}$. The selected microplastics in the study exhibit varying particle sizes, potentially leading to differing final results. Some microplastics may not be effectively separated from the brine channels, and there may exist

brine channels with even smaller widths. These factors have a significant impact on the separation of brine and ice, as well as the ultimate outcomes of the study.

We agree that MPs may be trapped in brine channels that are narrower than the MP. This is one hypothesis that we provide for the reduced concentration of small MPs in brine compared to large MPs in lines 294 to 300.

5. What is the significance of separating brine from ice, considering that in nature, brine and ice exist as an integrated entity? The environmental implications of centrifuging brine are unclear, and the separated brine only represents a portion of the total.

In this study, we wanted to assess the distribution of contaminants across multiple compartments in order to i) quantify distribution of contaminants between ice and brine and ii) to assess the mechanisms that drive contaminant fate across the system. Therefore, i) the brine and ice were separated to analyze the distribution of contaminants within the sea ice. ii) The ratio of concentrations between brine and ice reflects what happens at the freezing front and allows us to understand the incorporation mechanisms of contaminants. Also, the brine fraction of sea ice is where sympagic organisms thrive due to the availability of nutrients, the space and (sometimes) incoming sunlight. Consequently, the partitioning of contaminants between the brine and ice compartments are an important consideration when considering exposure. We have explained this in the manuscript:

Lines 238 - 245: To understand the mechanisms of contaminant incorporation in sea ice, we assessed the distribution of contaminants across the different sea ice phases. To do so, brine and ice were separated to analyse the distribution of contaminants within the sea ice. The ratio of concentrations between brine and ice was measured as it reflects what happens at the freezing front and allows us to understand contaminants' incorporation mechanisms. Also, the brine fraction of sea ice is where sympagic organisms thrive due to the availability of nutrients, the space and (sometimes) incoming sunlight. Consequently, the partitioning of contaminants between brine and ice are an important consideration when assessing exposure.

6. Figure 4 is missing. The text and figures do not match.

We thank the reviewer for catching this mistake. Figure 4 was accidentally labelled Figure 3, which has now been amended in the revised version of the manuscript.

7. Lines 230-232: Microplastics of small particle sizes may not all be able to pass through brine channels, and they may adhere to the edges of brine channels and ice. Additionally, aggregation of small-sized microplastics due to unstable freezing, which leads to an increase in relative particle size, may also trap them within brine channels.

We agree that MPs may be imperfectly drained out of the sea ice due to the brine channels tortuosity. We have discussed this as an explicit limitation in section 2.5.

Lines 395 – 401: Finally, this experimental design does not enable in-situ measurement of the adsorption of contaminants onto the ice or the aggregation state of colloids and particles. Therefore, processes such as particle and colloid adsorption by hydrophobic or van der Waals attraction, or entrapment of aggregates in the brine channels cannot be observed. Future experimental developments that enable in-situ quantification and

observation of colloids and particles in ice would allow to better quantify them and assess their surface affinity for ice and their aggregation state in-situ.

8. When discussing the occurrence concentrations of different types of microplastics, particle size and density are crucial factors to consider. Additionally, zeta potential and contact angle are also key characteristics for describing the properties of plastics. A more comprehensive discussion of the mechanisms should be included when analyzing the distribution of microplastics in ice, brine, and water.

While our work demonstrates the impact of contaminants' size and density in their incorporation, the role of contaminant adsorption onto the surface of the ice could not be studied due to the methods used. However, it is not a confounding factor to interpret the effect of size or density, since the smallest species are all hydrophilic (i.e., salts and rose bengal), and for the largest species (MPs), both hydrophilic (PET) and hydrophobic particles were used (PP) (see: quantification of contact angle by Schefer et al.⁴). For each of these polymers, the hydrophobicity does not change with size and the effect of size is dominant, as it controls the sinking/rising speeds (cf: Figures 4, 5, 6 and Supplementary Figure 5). Furthermore, at this ionic strength ($\cong 600$ mmol/L) the effect of electrostatic repulsion is nearly absent for all particles, regardless of their initial surface charge. Indeed, the zeta-potential, shown in Figure S6, is higher than -15 mV in brine and liquid.

We discuss the inherent limitations in our experimental design in Lines 395 to 401 (see previous response).

9. In Figure 3, the data related to polypropylene (PP) exhibit significant errors. For instance, the data points in brine deviate substantially from the main data cluster, suggesting the possibility of man-made error rather than systematic error. It is advisable to repeat the experiments to eliminate the large man-made error.

Experiments were performed by the same operator, and clear records were taken of the experimental and analytical conditions. This leads us to believe there are no fundamental errors when executing the experiments for PP compared to other contaminants. Rather, we believe that the deviation of the PP-MP represents the inherently more variable behavior of this material. As mentioned in lines 275 to 277, the removal of the top 0.5 cm ice section from the analysis of particle distribution (due to its affinity for the air water interface) may have contributed to a higher variability in PP-MPs concentrations.

Reviewer #2 (Remarks to the Author):

Review of: Understanding Contaminant Incorporation in Sea Ice: Impact of Size and Density, by Pradel et al.

Max Thomas

Pradel et al. present experimental results from a new sea-ice growth facility. A suite of contaminants with a range of physico-chemical properties were added to the experimental system, and their concentrations were measured in the underlying liquid, brine centrifuged from sea-ice samples, and in the sea ice remaining after centrifugation. Pradel et al. then interpret the concentration measurements based on the range of properties in the contaminants.

The experiments and measurements look competently done and are presented well. The choice of contaminants seems very well thought out and allows for a detailed and novel analysis of the mechanisms behind the behaviour of these chemicals. Sections 2.2 – 2.4 are particular highlights which have potential to be highly influential on future work. The work is original, and provides important, incremental knowledge gains for the sea-ice contaminant and process modelling community.

There is some work to be done on improving clarity, and to explicitly state how (unavoidable) biases and limitations in the measurements impact the conclusions. I was also confused by the lack of any bulk sea ice values being presented (the only major comment, which would require some more data analysis) which would help comparison with other work and might help simplify and strengthen conclusions.

I recommend the work for publication after the revisions listed below, which are fairly minor.

We thank the reviewer for his overall positive comments, especially for the recognition both of our novel experimental design and selection of the contaminant suite which we chose to study. We are specifically grateful that he finds the sections 2.2 – 2.4 to be of high interest and help to steer future research objectives, and indeed we are hopeful that through our own efforts or those of other scientists working in this field further advancements can be made with respect to contaminant fate in sea ice. Finally, it is notable that the reviewer has chosen to identify himself in the review (thank you!), which is not often done but helps us to better place the comments and suggestions we received into better context. We are confident that the revisions suggested have allowed us to improve our manuscript and to have even greater impact for future readers.

Major comments:

1) Line 22 – I don't think this statement is justified: 'While... particulate contaminants were enriched [or depleted] in sea ice'. Normally, 'sea ice' would here refer to the bulk value, which would be the concentration measured in melted ice and brine together. Bulk values aren't presented. I presume figure 3 is evidence to support this statement but I don't understand how the results link back to the statement, as a sea ice measurement is not presented. I suggest calculating 'sea ice' using the measured concentrations of a contaminant in ice and brine and the measured mass or volume of ice and brine. Possibly evidence is there already for this statement, and if so that needs to be made clear.

We have added the concentration of each contaminant in the bulk ice (ice + brine) to support this statement in Figure 4 and Supplementary Figure 5 along with the corresponding discussion.

Lines 200 - 232: First, the incorporation of each contaminant was assessed by comparing their concentration (C) in each the bulk ice ($C_{\text{bulk ice}}$ which corresponds to brine and ice) normalized by their initial concentration in the column before freezing (C_0) (Fig. 4). As mentioned above, the ice bulk salinity was reduced to 0.44 ± 0.07 of the initial salinity. Most other contaminants are depleted from the bulk ice, albeit in varying concentrations. The model dissolved contaminant, Rose Bengal, and the model colloidal contaminant, NP, were depleted in statistically similar proportions as sea salts (with $C_{\text{bulk ice}}/C_0 = 0.43 \pm 0.01$ and 0.38 ± 0.09 , respectively). This was followed by the other colloidal contaminant, Nanosoot, which was depleted to 0.48 ± 0.01 of its initial concentration.

The fate of particulate contaminants, was assessed by comparing high-density MPs and low-density MPs of two different size-classes resulting in four different MPs. High-density MPs were composed of polyethylene terephthalate (PET-MPs; 1.39 g cm^{-3}) and low-density MPs composed of polypropylene (PP-MPs; 0.91 g cm^{-3}). All MPs were fragments produced by cryo-milling and sieving, resulting in area equivalent diameter of $35.6 \pm 14.4 \text{ }\mu\text{m}$ and $45.7 \pm 20.7 \text{ }\mu\text{m}$ for small PET-MPs and PP-MPs, and $90.0 \pm 29.8 \text{ }\mu\text{m}$ and $72.3 \pm 37.2 \text{ }\mu\text{m}$ for large PET-MPs and PP-MPs, respectively. When inserting PP-MPs, many particles remained at air-liquid interface due to their light weight and hydrophobicity. These PP-MPs did not interact with the advancing ice front as would be the case during bottom freezing in the environment. Therefore, this fraction of PP-MPs was removed from the evaluation of distribution in the column (Figures 4, 5 and 6) but still considered when establishing the MPs recovery rate across the experiment (cf: Materials and Methods). When considering the portion of PP-MPs in the top layer of bulk ice, strong enrichment is observed as shown in Supplementary Figure 5 ($C_{\text{bulk ice}}/C_0 = 7.37 \pm 2.53$ and 3.22 ± 1.31 for Large and Small PP-MPs, respectively).

Overall, clear density-dependent trends are observed, even when excluding the top layer of ice with PP-MPs that could have been adsorbed to the air/water interface (Fig. 4) The high-density PET-MPs are depleted and the low-density PP-MPs concentrations remain almost constant. Levels of depletion/enrichment increase with particle size (Supplementary Section 1). The Large PET-MPs were significantly more depleted from the bulk ice than Small PET-MPs with $C_{\text{bulk ice}}/C_0 = 0.06 \pm 0.05$ and 0.49 ± 0.25 , respectively. This is explained by the fact that particle's settling or rising velocity is directly proportional to the power of its size. Conversely, Large PP-MPs had a higher $C_{\text{bulk ice}}/C_0$ value of 1.21 ± 0.59 than Small PP-MPs (0.86 ± 0.25).

Minor comments:

1) Line 12 – Sea ice forms near human activity too (Canadian towns, Longyearbyen) so please remove 'despite being far from human activity'. You may want to add that it transports contaminants as this is important and mentioned prominently in the intro.

Yes, we agree with the reviewer and have now removed this statement about remoteness and only mention its capacity to transport contaminants.

Lines 12 – 13 : Sea ice accumulates various contaminants and can redistribute them laterally as the ice drifts and vertically as ice melts.

2) Line 14 – Please make clear what aspects are 'novel'. The suite of chemicals is novel, but the cold plate, growth in tubes, artificial salt solution are not novel. If it is some combination of the design choices that is novel for some specific novel need, that should be made clear and justified. The methods don't need to be novel for the work to be novel and worthwhile.

We can understand the reviewers concerns that many researchers over emphasize the novelty of their work, and a combination of factors is in itself not necessarily novel since there are many unique research lines which can be explored. There, we have more clearly specified what is novel about this experiment in the main text.

Lines 94 - 97: The experimental set-up was a novel development since i) it generated environmentally representative artificial sea ice by keeping the internal pressure and the salinity of the underlying water constant and ii) it allowed us to quantify model contaminants within the sea ice.

3) Line 18 – What precisely does ‘difficult’ mean? Are you referring to contamination issues?

We have modified the sentence in the abstract to clarify what we mean by difficulties in quantifying certain contaminants:

Lines 20-21: and difficulties quantifying certain contaminants due to the lack of standardized methods.

We have also detailed the reasons behind this difficulty in the introduction

Lines 74 - 78 : Furthermore, quantifying some contaminants, such as small MPs (<10 μm) and NPs, in environmental matrices is challenging due to the lack of standardized methods. Some persistent challenges, such as differentiating MPs and NPs from natural organic matter (NOM) and avoiding, contamination during sampling and analysis require time-intensive sampling and extraction methods. Overall, this limits the amount of samples which can be measured.

4) Line 20 – Are kDa and μm related to each other somehow? I don’t see how a range of values from x to y is useful when x and y have different units.

1 nm is roughly equivalent to 1 kDa. Daltons are the more scientifically accurate unit to assess the size of small colloids, since they are malleable and their dimensions will depend on solution chemistry. However, we agree that using two different units is confusing and so we will use the definition of colloids as 1nm to $1\mu\text{m}$ as per the definition of Lead and Wilkinson (2007). Changes were made throughout the text and the reference was added line 66.

5) Line 58 – The authors will need to be more clear about what ‘brine rejection’ is referring to. Generally, ‘brine rejection’ refers to brine moving from the sea ice to the ocean below, which desalinates the sea ice. This happens after sea ice has grown and incorporated everything. There is insignificant segregation at the advancing interface, so everything that was in the water is incorporated into the growing sea ice, then rejected (Notz & Worster, 2009, <https://doi.org/10.1029/2008JC004885>). What the authors are describing is salt and other dissolved species being excluded from the ice matrix such that they are partitioned to the brine, but remain within the sea ice.

Thank you for bringing this mistake to our attention. We have modified this sentence and added another one, as well as the Notz and Worster reference, for further clarity.

Lines 60- 63: During bottom freezing, dissolved molecules such as sea salts are depleted from the bulk ice and accumulated in brine pockets and channels, a process known as salt segregation or partitioning. This can be followed by gravity drainage, or flushing of the brine in the sea ice which leads to overall desalination of the bulk sea ice.

6) Line 59 – The authors should be careful to use ‘ice’ to mean the frozen, ice matrix of sea ice,

and 'sea ice' or 'bulk sea ice' to mean the collection of ice and brine. Here, 'bulk ice' is used, which is confusing.

We have made the correction here and throughout the text.

7) Line 62 – (see minor comment 4)

We have made the correction.

Lines 66 and 67: dissolved (< 1nm) [...] colloidal (1nm to 1 µm)

8) Line 77 – 'Particulate species with low densities are expected to be enriched...' Enriched relative to what? Where does this hypothesis come from? The other hypotheses follow from the introduction, but this one seems unconnected to the previous introduction material.

We have added some clarifications to this part :

Lines 85 – 88 : On the other hand, particulate species with low densities are expected to be enriched in the ice relative to the underlying water due to their positive buoyancy which moves them towards the advancing ice front. Conversely high-density particles, due to their negative buoyancy, are expected to be depleted by sinking below the advancing ice front.

9) Line 83 – This is probably the place to address minor comment 2.

We have addressed this comment :

Lines 94 to 97: The experimental set-up was a novel development since i) it generated environmentally representative artificial sea ice by keeping the internal pressure and the salinity of the underlying water constant and ii) it allowed us to quantify model contaminants within the sea ice.

10) Line 90 – Defining ice and brine earlier could solve previous comments.

We have added the distinction between ice and bulk ice, lines 58 and 61. We have also added some clarity to this sentence :

Lines 100 – 102 : The artificial sea ice formed was harvested to characterize the artificial sea ice structure and to quantify species in the underlying liquid, the bulk ice, as well as its two fractions: the solid ice matrix (hereafter called ice) and the brine.

11) Line 92 – 'Trapped' probably isn't the right word here.

We have changed the term to "incorporated" (line 103)

12) Line 92 – 'Collectively this' is imprecise (and should be 'these' I think). I would suggest replacing this with 'Our results' or 'These knowledge gains', if that's what is meant.

Thank you for your suggestion. We have made the suggested changes.

Line 103-104 : Our results improve our ability to [...].

13) '4' is repeated in the figure 1 caption and '2' is missing. Also 'column' should be 'a column' or 'columns'. This figure is very clear and informative.

We made the appropriate corrections in the caption of Figure 1.

14) Line 108 – 'vertical temperature gradient'.

We added "vertical" to "temperature gradient" in line 119.

15) Line 114 – 'average' to 'mean' (I presume).

We have made corrections to "mean" throughout the text.

16) Line 129 – 'brine salinity approximately doubled'. The authors should calculate and report to temperature in Vancoppenolle et al. (2019). This comparison will help the authors think about how representative their centrifuged brine is of the in situ brine.

We have added this comparison to our discussion

Lines 142 to 146 : This brine salinity was compared to predictions of brine salinity as a function of temperature⁵. We used -1.7 °C as the temperature at the bottom of the ice core since it is the freezing temperature of our artificial seawater (31.28 g L⁻¹) and the temperature of the cooling element, -5.1 °C, as the temperature at the top of the ice core. This suggests that the brine salinity should be 57.8 g L⁻¹ which is consistent with our observations.

17) Lines 133-135 – It is great that the authors have thought deeply about this aspect of the experiment and have noted it here. It then needs to feature in the discussion (see also point 2.4).

This aspect is now further discussed in lines 196 to 199. We have further elaborated on the limitations involved in separating brine by centrifugation throughout the manuscript

Lines 165 - 167: This can again be attributed to some brine loss, as well as some residual brine pockets within the ice that were not connected to the bottom of the ice, and therefore not drained by centrifugation. Despite these experimental limitations, our results agree well with the properties of natural sea ice.

Lines 261 to 264 : Note that, as for the salts, it is also possible that some Rose Bengal in brine pockets was not centrifugated out of the ice. This would increase their measured concentration in the ice and decrease their measured concentration in the brine.

18) Lines 136-143 – These values are reassuring, and suggest the artificial sea ice is a reasonable approximation of the real world. I would note that the sea ice in this study is very young (<5 cm), whereas real world sea ice is considered 'young' while less than 30 cm – I would expect 30 cm sea ice to have lower salinity and solid fraction. Also, the values reported by the authors here will be biased by brine loss during sampling (so before centrifuging). This bias is practically unavoidable. These caveats should be noted here. The authors could compare to Notz & Worster (2008) which present in situ measurements of bulk salinity and solid fraction (<https://doi.org/10.1029/2007JC004333>), and have measurements in <5 cm sea ice. For

comparable thicknesses the authors will see the in situ measurements will have a higher bulk salinity and solid fraction.

Thank you for bringing this to our attention. We have further discussed these aspects in the main manuscript.

Lines 161 to 170 : For example, Notz and Worster (2008) observed that sea ice of comparable thickness had a bulk salinity ranging from 15 to 25 g L⁻¹ ⁶. Our lower bulk salinity may be explained by the fact that brine may be lost when the artificial sea ice is pulled out of the column. The solid mass fraction of their sea ice ranged from 0.5 to 0.7⁶ which is lower than our solid mass fraction of 0.85 ± 0.04 . This can be again attributed to some brine loss, as well as some residual brine pockets within the ice that were not connected to the bottom of the ice, and therefore not drained by centrifugation. Despite these experimental limitations, our results agree well with the properties of natural sea ice. It should also be noted that this artificial sea ice produced represents the very early stages of sea ice growth, since young sea ice can reach up to 30 cm depth and can desalinate to 4 to 6 g L⁻¹ ^{1,6}.

19) Line 155 – Why did the freezing element have a curved shape?

To avoid any entrapment of air bubbles between the freezing element and the surface of water, the freezing elements was designed with a curved shape. This allowed the air to escape when the freezing element was inserted.

Lines 184 - 186 : This curved shape was used to allow air bubbles to escape as the cooling element was plunged into the water, to enable full contact between cooling the element and the water.

20) Figure 2b – Could the authors please annotate the image? I had to guess what the lenticular structures were. Also, what is 'black' in the image?

We have now annotated the image in order to make the areas of interest more immediately visible. What appears "black" is simply the edges of the grey zones which delimit the ice. Indeed, in the rendering the grey area has darker edges at the border with the white areas which show the "air" (i.e.: drained brine).

21) Line 171 – 'if' to 'is'

We have made the change in Line 230.

22) Line 229 – Needs an 'and' between the reported values.

We have made the change in Line 289.

23) Line 255 – 'already' to 'already been'

We have made the change in Line 314.

24) Section 2.4 – As a minimum, I would like to see a line here reminding the reader that Cice in this study includes brine in isolated pockets. It would also be very useful to get the authors take on

how the inclusion of isolate brine within Cice impacts their conclusions. I found it difficult to understand how, for example, the exposure of microorganisms to contaminants is impacted by having this imperfect separation of phases in the measurement. The authors should also note bias from brine loss upon sampling and speculate as to impacts on conclusions. Possibly these caveats could form a short paragraph, as this paragraph is already quite complex and well put together.

We have discussed the limitations of our experiments and how these affect our interpretations of in section 2.5 called “2.5. Suitability of lab-scale experiments to study incorporation of contaminants in sea ice”. Specifically, this point was addressed in the following way:

Lines 387 to 395: With respect to the type observations that can be made, this design does not enable the use of methods that avoid the loss of brine as it the sea ice is removed (i.e.: by inserting a box under the sea ice as it is pulled out⁵⁴). Furthermore, the use of centrifugation may not drain the brine pockets that are not connected to the bottom of the ice core. Therefore, the brine content is underestimated and the ice salinity is overestimated, compared to methods which measure brine content in-situ⁴³. Future experiments could incorporate in-situ temperature and salinity measurements to record the ice growth and brine content in real time. However, since microorganisms also thrive in the brine channels that are connected to the underlying water, the brine fraction studied here is the more biologically-active fraction of the sea ice.

25) Figure 5 is excellent.

Thank you !

26) Line 323 – ‘pressure of crystal growth’? Do the authors mean actual pressure (as in Pa), and if not, what?

We have clarified this sentence to mean the fact that ice solidification can move colloids and push them towards each other which increases the possibility of interaction.

Lines 427- 431: Indeed, sea ice is an ideal environment for the aggregation of colloids and potentially also for the self-assembly of dissolved species into colloidal or particulate species due to i) the local increase in salt and colloid concentrations in brine, and to ii) the growth of solid ice which can push colloids closer together thereby increasing the possibility of interaction.

27) Glossary: The statements about this sea ice containing no gas should be softened. Gas is often found in young sea ice, and there could be some here, unless the authors have some evidence to the contrary.

We have made the changes accordingly:

Lines 597 - 598: The artificial sea ice created here contains solid and liquid fractions (i.e., ice and brine, respectively). Gas bubbles were not observed by μ -CT.

28) At the end of the paper I realised I don't understand why the low density MP were so depleted in the liquid. Scanning back through I couldn't find the answer. Could the authors please explain this, or make the explanation more prominent.

We have made this explanation more prominent:

- 1) The first part of section 2.2 (lines 210 to 232) discusses the incorporation of MPs in bulk ice, and more clearly demonstrates the impact of MP density
- 2) The paragraph which compares MP concentrations in ice and liquid states more prominently the role of particle density.

Lines 266 to 281: As demonstrated with the enrichment/depletion trends of MPs in bulk ice, MP's enrichment/depletion patterns are attributed to i) polymer density, which confer a positive buoyancy (i.e., rising) for the low-density PP-MPs and a negative buoyancy (i.e., settling) for the high-density PET-MPs and to ii) size, which accelerates the enrichment/depletion of MPs (Fig. 4, Fig. S5, and Supplementary Section 1). This is further demonstrated by comparing concentrations in ice and in the underlying liquid. [...] PP-MPs' enrichment was compensated by a depletion in the underlying liquid for the PP-MPs ($C/C_0 = 0.308 \pm 0.308$ and 0.682 ± 0.158 for large and small PP-MPs, respectively).

29) I would also suggest saying 'The impact of...' in the title as it's plainer English. (Just a note to say I thought the section titles were very helpful. The declarative statements made the paper easy to follow.)

We have made the suggested change to the title 😊

Reviewer #3 (Remarks to the Author):

Understanding Contaminant Incorporation in Sea Ice: Impact of Size and Density
Pradel et al

This manuscript details the experimental set-up and application of an artificial sea ice chamber system that allows the intimate study of solutes and particles, present in sea water, and their interaction with freshly forming ice. The authors are to be applauded for their relatively simple but elegant experimental set up. To my knowledge this small-scale system with multiple plastic polymer columns, an air-side cooling system (representing real Polar conditions over surface seawater) combined with a pragmatic pressure release tube is novel. Given the chemical and physical measurements of the fresh ice plug (with CT imaging) then similarities to initial ice in the Arctic is convincing. Hence this column system does provide a unique system to study the uptake, release and fate of both dissolved and particulate contaminants in ice (e.g. persistent organic chemicals, plastic particles, POC/DOC, heavy metals etc). The authors then demonstrate the utility of the apparatus by examining the fate of selected solutes, colloids and plastic particles (varying density) during a freeze experiment. Enrichment/depletion of these species relative to their occurrence in seawater is then reported with comparisons, where possible, to field and laboratory studies. The study is at the standard for publication however there are several issues that the authors need to comment on or address before publication could proceed.

We thank the reviewer for their enthusiastic and positive comments for our work, and we hope that this experimental design can also be implemented by others in the future in order to accurately assesses various processes and contaminant pathways within sea ice. With the additional changes

suggested by this and other reviewers, we believe that we have further refined the manuscript for future readers to give the work its full potential.

1) The columns and their seawater contents were agitated by being placed on an orbital shaker during the freeze period. It is not entirely clear why this was undertaken but does make sense in that particulate matter is likely to remain suspended in the water column, rather than deposited to the vessel floor (for the higher density particles). This agitation however does not represent a sub-ice seawater current though. This could be achieved using a small submerged pump or similar, which is common in larger sea ice chambers, which hence do resemble the surface ocean with this regard. On this note there doesn't appear to be incorporation of sensors to measure physical parameters of the seawater itself. Most notably the temperature/salinity profile down the water column should be essential. For example, the expulsion of brine with ice aging or the release of low salinity melt water during ice melt, may result in stratified water layers which will affect chemical/material flows and yet are not easily distinguished in the current system.

As the reviewer has already identified, the purpose of placing the columns on orbital shakers was to aid in homogeneously dispersing the contaminants, particularly the MPs. We agree that circular agitation motion is not representative of underwater currents, but despite this limitation, we were able to achieve and structure in the sea ice which is comparable to very young natural sea ice. It is also notable that the dissipation energy generated through the orbital shaker is on the order of $10^{-6} \text{ m}^2 \text{ s}^{-3}$ which is a median value for the open ocean, according to Arnott et al (2021)⁷.

To generate currents that are more environmentally representative and to measure the temperature and salinity profiles of the ice and of the underlying seawater would require making a larger experimental set-up. Indeed, a lateral current cannot easily be generated in a column of 6.5 cm diameter by 44 cm height. Also, we had put temperature sensors throughout the sides of one column but the temperatures were too strongly influenced by the ambient temperature since the sensors could not be entirely plunged in the column due to space constraints. So, small sensors plunged into a larger volume of ice would enable in-situ measurements of temperature and salinity while minimizing disturbance.

We have addressed these experimental limitations in Section 2.5 “3. Suitability of lab-scale experiments to study sea ice processes.”

Lines 378 – 382 : With respect to the type of ice grown, the advantage of our experimental design is that, by being relatively small it allows many replicates to be performed since the freezing time and the amount of model contaminants required are reduced. However, this smaller size does not enable the generation of underwater currents which occur naturally under sea ice. Despite this, the artificial sea ice formed is similar to natural ice.

Lines 392 – 393 : Future experiments could incorporate in-situ temperature and salinity measurements to record the ice growth and brine content in real time.

2) The freezing process using the aluminium cooling vessel will interfere with air side transfer of particles and vapour/gaseous species. While the focus is clearly on studying the ice itself and the ice-seawater interface, this is nonetheless a significant oversight particularly for studying the transfer of material/chemicals across the ice-air interface. Importantly, the introduction of material to the surface of the ice to simulate ice-rafted snow for example cannot be undertaken with the current set up.

Indeed, our experimental set-up neglects the processes occurring at the air/ice interface and focuses solely on processes occurring at the water/ice interface. This is due to our choice of looking specifically at bottom freezing, which, as indicated in the introduction, forms most of the volume of sea ice (lines 58 to 60). As noted by Hall et al. (2023)⁸, experimental designs which can generate both frazil and columnar ice generally require larger volumes. However, this also has limitations such as reduced capacity for replicate experiments and reduced control on temperature gradients. In sum, since no experimental design is perfect, we have discussed the limitations and how they impacts our results in section 2.5.

Lines 382 to 385: One must also keep in mind that this experimental set-up generated columnar ice, which grows below the layer of frazil ice and, as such, does not interact with contaminants at the air/water interface. However, this columnar ice is most common in the Arctic, which receives more anthropogenic contaminants than the Antarctic.

Lines 422 – 423: This can help to understand whether contaminant incorporation occurs during basal ice growth or frazil ice growth.

3) In the Introduction the authors cite those studies that have observed microplastics in Polar sea ice at concentrations orders of magnitude higher than concentrations present in the underlying sea water. This is one of the drivers behind experimental simulations of the seawater/sea ice system like this study. However, in the experimental work for the microplastics the authors discarded the uppermost 0.5cm ice layer on the basis that lower density particles had accumulated in the water-air interface and hence had become artificially rafted onto the ice-surface. Surely, this is the very process which will be occurring to some extent for positively-buoyant plastic particles in the real marine environment! Does this ice-rafting of some types of microplastics occur during early formation of grease/frazil ice?? This is a key question which hopefully this new experiment apparatus could investigate, but which the authors have viewed as an artifact!!

We agree that the early stages of sea ice growth (i.e., grease/frazil ice) may incorporate MPs that are present at the seawater/air. However, the type of artificial sea ice produced here is not grease or frazil ice, but columnar ice that grows by bottom freezing and does not interact with the air. This is why we have discarded the MPs stuck at the air/ice interface (cf: previous response). We would like to point out that we have another study, currently under review, which uses a similar experimental set-up to generate frazil ice and to investigate this specific process. So stay tuned for more information on this topic in (hopefully!) the near future! ☺

P1, line 28 contaminantation

Line 31 : Contamination

P2, line 72in [a] controlled environment...

Line 83 : in a controlled

P2, line 94will be exposed to.

Line 110 : will be exposed to

References

1. Thomas, D. N. & Dieckmann, G. S. *Sea Ice*. (Blackwell Publishing Ltd, 2010).
2. IOC, SCOR & IAPSO. *The International Thermodynamic Equation of Seawater – 2010: Calculation and Use of Thermodynamic Properties*. 196 https://www.teos-10.org/pubs/TEOS-10_Manual.pdf (2010).
3. McDougall, T. J. & Barker, P. M. *Getting Started with TEOS-10 and the Gibbs Seawater (GSW) Oceanographic Toolbox*. (SCOR/IAPSO WG127, 2011).
4. Schefer, R. B., Armanious, A. & Mitrano, D. M. Eco-Corona Formation on Plastics: Adsorption of Dissolved Organic Matter to Pristine and Photochemically Weathered Polymer Surfaces. *Environ. Sci. Technol.* **57**, 14707–14716 (2023).
5. Vancoppenolle, M., Madec, G., Thomas, M. & McDougall, T. J. Thermodynamics of Sea Ice Phase Composition Revisited. *J. Geophys. Res. Oceans* **124**, 615–634 (2019).
6. Notz, D. & Worster, M. G. In situ measurements of the evolution of young sea ice. *J. Geophys. Res.* **113**, 2007JC004333 (2008).
7. Arnott, R. N., Cherif, M., Bryant, L. D. & Wain, D. J. Artificially generated turbulence: a review of phycollogical nanocosm, microcosm, and mesocosm experiments. *Hydrobiologia* **848**, 961–991 (2021).
8. Hall, B., Johnson, S., Thomas, M. & Rampai, T. Review of the design considerations for the laboratory growth of sea ice. *J. Glaciol.* 1–13 (2023) doi:10.1017/jog.2022.115.

Answer to reviewers

Reviewer #1 (Remarks to the Author):

As the authors have repeatedly stated, the simulation experiments indeed have many limitations, but they have made the greatest possible effort to address my comments. I recommend that this version of the manuscript can be published.

Thank you for your time reviewing our work.

Reviewer #2 (Remarks to the Author):

The authors have thoroughly responded to my first review, and I am happy that the manuscript is fit for publication.

There are three further points I suggest they address:

1) Comment 2, new lines 94-97: There are existing systems which keep pressure from building, so this aspect is not novel (Thomas et al. (2020) for example). The chemical suite should be mentioned here as that is novel. The curved cooling plate and net are also new to the sea-ice literature I think.

We have made the changes according to your feedback :

Lines 94-97: The experimental set-up was a novel development since i) it generated environmentally representative artificial sea ice by keeping the salinity of the underlying water constant, and ii) it allowed us to quantify a suite of model contaminants within the sea ice.

2) Figures 4 and 5 should be combined as they share axes and this would aid comparing them.

Thank you for your suggestion. However, we find that combining the two figures would make them too data-heavy and less clear to read. Furthermore, the distinction between bulk ice and (brine-free) ice might be blurred by combining them. Therefore, for clarity, we would prefer to keep them separate.

3) On lines 24-25 in the abstract, where the enrichment/depletion is discussed, it should be made clear that this is relative to salt.

This is relative to the initial concentrations in seawater. We have specified this :

Lines 24-26 : While high-density particulate contaminants were depleted in sea ice and low-density particulate contaminants were enriched relative to their initial concentration in seawater, both were engulfed and traveled in the wider brine channels.

Reviewer #3 (Remarks to the Author):

I have now reviewed the authors' responses to the comments made on the manuscript and they have satisfactorily addressed the points that I raised and as well the points raised by the other reviewers. Importantly the revised manuscript reflects this with substantial additions/changes made to the Results & Discussion. These additions improve the manuscript markedly and now highlight the limitations and/or biases that occur in the experimental ice columns relative to the real environment. The rationale for excluding or removing the ice rafted plastic particles is now clearly articulated. Overall I recommend that this manuscript can now be published.

Thank you for your time reviewing our work.

There are a couple of typos in the following lines...

377 "With respect to the type [OF] observations that can be made, this design does not enable the use

388 of methods that avoid the loss of brine as [it] the sea ice is removed"

We have corrected both typos.